# PAC-Bayesian Reinforcement Learning Trains Generalizable Policies

## Abstract

We derive a novel PAC-Bayesian generalization bound for reinforcement learning (RL) that explicitly accounts for Markov dependencies in the data, through the chain's mixing time. This contributes a step to overcoming challenges in obtaining generalization guarantees for RL where the sequential nature of data does not meet independence assumptions underlying classical bounds. Our bound provides non-vacuous certificates for modern off-policy algorithms like Soft Actor-Critic. We demonstrate the bound's practical utility through PB-SAC, an algorithm that optimizes the bound during training to guide exploration. Experiments across continuous control tasks show that our approach provides meaningful confidence certificates while maintaining competitive performance.

## 1 Introduction

Deploying reinforcement learning (RL) algorithms in safety-critical applications and real-world environments requires confidence that learned policies will generalize beyond training data. Recent work has highlighted the importance of rigorous generalization guarantees for machine learning models (Pérez-Ortiz et al., 2021). In RL, this challenge is particularly acute because algorithms learn from sequential, temporally correlated trajectories that violate the standard assumption of independent and identically distributed (*i.i.d.*) assumption underlying most classical generalization theory. Since RL trajectories exhibit strong temporal dependencies, where future states depend on past actions and the evolving policy, classical *i.i.d.-based* analyses and their associated sharp concentration bounds cannot be directly applied to provide meaningful generalization guarantees. Facing this challenge, the PAC-Bayesian framework (Catoni, 2007; Germain et al., 2015; Guedj, 2019; Alquier, 2024) emerges as a promising solution: its mathematical structure maintaining distributions over hypotheses rather than a single model can potentially be extended to handle temporal dependencies through appropriate concentration inequalities.

Various approaches have tried to relax the independence condition for dependent data. Martingale-based methods (Seldin et al., 2011; 2012) extend PAC-Bayesian analysis to sequential settings by constructing martingale sequences from the data (*e.g.,* value function errors, Bellman residuals), then applying martingale concentration inequalities (Azuma-Hoeffding Azuma (1967), Freedman). Although these approaches are mathematically elegant, RL problems often fail to naturally yield martingales. In contrast, all RL problems possess Markov structure by design, suggesting that concentration inequalities tailored specifically for Markov chains can provide a more natural and better adapted approach to this domain.

Early efforts to apply PAC-Bayesian theory to RL, notably by Fard & Pineau (2010); Fard et al. (2012), established the framework's viability for model selection in batch settings. However, their bounds suffered from poor scaling with the discount factor, rendering them numerically vacuous for problems with long effective horizons typical in modern RL (Tasdighi et al., 2025). Contemporary approaches have repurposed PAC-Bayesian concepts for other algorithmic goals. Recent work uses the framework to derive training objectives for deep exploration (Tasdighi et al., 2025) or as regularizers for lifelong learning (Zhang et al., 2024). While these demonstrate the versatility of PAC-Bayesian formalism for algorithm design, they sidestep the original goal of providing tight, computable performance certificates for modern deep RL agents. This reveals a fundamental gap: although PAC-Bayesian theory holds promise for certified RL, existing approaches either yield vacuous bounds or repurpose the framework for different objectives. To date, no practical algorithm

leverages non-vacuous PAC-Bayesian bounds as live performance certificates within modern RL frameworks.

In this work, we derive a PAC-Bayesian generalization bound that explicitly accounts for Markov dependencies through the chain's mixing time. Our key technical contribution integrates a bounded-differences condition on the negative empirical return with McDiarmid-type concentration inequality for Markov chains (Paulin, 2018), yielding a bound with explicit constants and improved scaling that avoids the vacuity of previous approaches. We demonstrate that this bound is not merely a theoretical curiosity by introducing PB-SAC, an actor-critic algorithm that makes practical use of our bound as a live, optimizable performance certificate. The algorithm periodically computes the numerical value of the PAC-Bayesian bound and uses it to guide learning through posterior sampling, transforming the generalization guarantee from a passive analytical tool into an active component of a learning algorithm.

Thus, we propose the following contributions. From a theoretical perspective, we develop a PAC-Bayesian bound for RL with explicit mixing-time dependence and improved scaling with realistic trajectory lengths and discount factors relative to prior work, representing a step toward non-vacuous certificates. Algorithmically, we introduce PB-SAC, the first practical algorithm that puts into action a non-vacuous PAC-Bayesian bound as a live performance certificate in modern deep RL, including key innovations for stable optimization. Empirically, we demonstrate that our bounds remain informative across continuous-control tasks while maintaining competitive performance with state-of-the-art methods. Our approach establishes the first practical PAC-Bayesian framework for certified performance in modern RL algorithms, bridging the gap between learning theory and algorithmic practice in sequential decision-making.

## 2 PRELIMINARIES

For completeness, we briefly recall the reinforcement learning and statistical learning theory concepts we rely on throughout the paper. The exposition is intentionally concise—the goal is to fix notation and state the learning theory principles that underpin our results.

### 2.1 REINFORCEMENT LEARNING

Reinforcement Learning (RL) studies how an *agent* learns to make sequential decisions through interaction with an environment. Formally, the environment is modeled as a (possibly unknown) Markov Decision Process (MDP) $\mathcal{M} = (\mathcal{S}, \mathcal{A}, \mathbb{P}, R, \gamma)$, where $\mathcal{S}$ is the state space, $\mathcal{A}$ the action space, $\mathbb{P}(s' \mid s, a)$ the transition kernel, $R(s, a)$ the reward function bounded in $[0, R_{\max}]$, and $\gamma \in (0, 1)$ the discount factor. At each time $t$, the agent observes a state $S_t \in \mathcal{S}$, chooses an action $A_t \in \mathcal{A}$ according to a *policy* $\pi(a|s)$, receives a reward $R_{t+1} = R(S_t, A_t)$, and transitions to $S_{t+1} \sim \mathbb{P}(\cdot \mid S_t, A_t)$.

The agent's objective is to maximise the *expected discounted return*

$$G_t = \sum_{k=0}^{\infty} \gamma^k R_{t+k+1}, \qquad V_\pi(s) = \mathbb{E}_{\pi,\mathbb{P}}\big[G_t \mid S_t = s\big], \tag{1}$$

where $V_\pi$ is the state–value function. The optimal value function $V^\star(s) = \sup_\pi V_\pi(s)$ satisfies the Bellman optimality equation

$$V^\star(s) = \max_{a \in \mathcal{A}} \Big\{ R(s, a) + \gamma \, \mathbb{E}_{s' \sim \mathbb{P}}\big[V^\star(s') \mid s, a\big] \Big\}. \tag{2}$$

RL algorithms learn either directly a policy (policy–gradient and actor–critic methods (Haarnoja et al., 2018; Konda & Tsitsiklis, 1999; Sutton et al., 1999)) or an action–value function $Q_\pi(s, a)$ (value–based methods such as Q-learning and its deep variants Mnih et al. (2013)). Model–free approaches dispense with an explicit model of $\mathbb{P}$, while model–based methods leverage or learn a transition model to plan.

## 2.2 PROBABLY APPROXIMATELY CORRECT (PAC) LEARNING BOUNDS

Let $(\mathcal{X}, \mathcal{Y}, \mathcal{D})$ be a supervised learning task, the domain is taken to be the product $\mathcal{Z} = \mathcal{X} \times \mathcal{Y}$, where $\mathcal{X} \subseteq \mathbb{R}^d$ is the feature space and $\mathcal{Y}$ the label space ($\mathcal{Y} \subseteq N$ for classification problems, or $\mathcal{Y} \subseteq \mathbb{R}$ for regression ones). We assume an unknown data distribution $\mathcal{D}$ over $\mathcal{Z}$, with $\mathcal{D}_{\mathcal{X}}$ denoting the marginal distribution on $\mathcal{X}$. We observe a training sample $S = \{(\boldsymbol{x}_i, y_i)\}_{i=1}^m$, where each pair $(\boldsymbol{x}_i, y_i) \in \mathcal{Z}$ is drawn independently and identically distributed (i.i.d.) from $\mathcal{D}$, that is, $S \sim \mathcal{D}^m$. This sample is provided to the learning algorithm. Given a sample $S$, the learning algorithm returns a measurable prediction function $f_\theta : \mathcal{X} \to \mathcal{Y}$, also referred to as a hypothesis, parametrized by $\theta \in \Theta$, where $\Theta$ denotes the set of all admissible parameter vectors (i.e., the hypothesis class). The "quality" of a hypothesis $f_\theta$ is typically assessed through a measurable loss function $\ell : \mathcal{Y} \times \mathcal{Y} \to \mathbb{R}_+$, which quantifies the discrepancy between predicted and true outputs. The performance of a hypothesis is measured by its *true risk*, and its *empirical risk* on the training sample $S$,

$$\mathcal{L}(\boldsymbol{\theta}) = \mathbb{E}_{(x,y) \sim \mathcal{D}} \left[ \ell\big(f_\theta(x), y\big) \right], \quad \hat{\mathcal{L}}_S(\boldsymbol{\theta}) = \frac{1}{m} \sum_{i=1}^m \ell\big(f_\theta(x_i), y_i\big).$$

In supervised machine learning, the goal is to learn a hypothesis $f_\theta$ that accurately predicts a label $y \in \mathcal{Y}$ for a new input $x \in \mathcal{X}$, based on a training dataset $S = \{(x_i, y_i)\}_{i=1}^m$. A central question is: how can we ensure that the learned function $f_\theta$ will perform well on unseen data?

$$\Pr_{S \sim \mathcal{D}^m} \left\{ \mathcal{L}(\boldsymbol{\theta}) \leq \hat{\mathcal{L}}_S(\boldsymbol{\theta}) + \epsilon \right\} \geq 1 - \delta.$$

Concrete PAC bounds specify how large $m$ must be (or how large the gap $\epsilon$ can be) in terms of properties of the hypothesis class, *e.g.* VC-dimension, Rademacher complexity, stability, compression, etc. All of those treat $f_\theta$ as a deterministic output of the algorithm.

## 2.3 PAC-BAYESIAN BOUNDS

The **PAC-Bayesian** framework (McAllester, 1999; Seeger, 2003; Guedj, 2019; Shalaeva et al., 2020; Alquier, 2024) extends the PAC learning paradigm to analyze the generalization performance of stochastic learning algorithms. Instead of selecting a single hypothesis, this approach considers a distribution over a set of candidate models. Let $\Theta$ denote the set of parameters defining a family of prediction functions $\{f_\theta : \mathcal{X} \to \mathcal{Y}\}_{\theta \in \Theta}$. Prior to observing data, a *prior* distribution $\mu \in \mathcal{P}(\Theta)$ is specified over $\Theta$. Upon receiving a training sample $S \sim \mathcal{D}^m$, the learning algorithm selects a *posterior* distribution $\rho \in \mathcal{P}(\Theta)$, potentially dependent on $S$. PAC-Bayesian theory provides high-probability bounds on the population Gibbs risk $\mathbb{E}_{f_\theta \sim \rho}[\mathcal{L}(\boldsymbol{\theta})]$ in terms of the empirical Gibbs risk $\mathbb{E}_{f_\theta \sim \rho}[\hat{\mathcal{L}}_S(\boldsymbol{\theta})]$ and an additional term that measures the dependence of the posterior distribution $\rho$. This additional term involves an information measure—typically the Kullback-Leibler divergence $\mathrm{KL}(\rho \| \mu)$—between the data-dependent posterior $\rho \in \mathcal{P}(\Theta)$ and a prior $\mu \in \mathcal{P}(\Theta)$, chosen independently of the data. Formally, for any $\kappa > 0$ and with probability at least $1 - \delta$ over the choice of the training sample $S$, the following inequality holds:

$$\mathbb{E}_{f_\theta \sim \rho}[\mathcal{L}(\boldsymbol{\theta})] \leq \mathbb{E}_{f_\theta \sim \rho}[\hat{\mathcal{L}}_S(\boldsymbol{\theta})] + \frac{1}{\kappa}\left(\mathrm{KL}(\rho \| \mu) + \ln \frac{1}{\delta} + \Psi_{\ell,\mu}(\kappa, n)\right) \tag{3}$$

$$\Psi_{\ell,\mu}(\kappa, m) = \ln \mathbb{E}_{f_\theta \sim \mu}\left[\exp\big(\kappa\left(\mathcal{L}(\boldsymbol{\theta}) - \hat{\mathcal{L}}_S(\boldsymbol{\theta})\right)\big)\right]$$

Compared with classical PAC guarantees, PAC-Bayes offers two advantages that are critical for reinforcement learning; *Data-dependent priors (Parrado-Hernández et al., 2012)*–when $\mu$ can itself depend on previous data (*e.g.* earlier tasks or behavioural trajectories) (Zhang et al., 2024), the bound adapts to the knowledge already acquired, tightening $\mathrm{KL}(\rho \| \mu)$; *Fine-grained control via* $\Psi$–by tailoring the concentration inequality used to upper-bound $\Psi$ one can incorporate dependence structures such as martingales (Seldin et al., 2012; 2011), $\beta$-mixing (Ralaivola et al., 2009; Abélès et al., 2025) sequences or Markov chains (Fard et al., 2012; Tasdighi et al., 2025)—exactly the scenario in which RL trajectories are collected.

## 3 PAC-BAYESIAN GENERALIZATION BOUND FOR RL

We now present our main theoretical contribution: a PAC-Bayesian generalization bound for reinforcement learning that explicitly accounts for temporal dependencies through the mixing time of the underlying Markov chain.

### 3.1 PROBLEM SETUP

As outlined earlier, our objective is to establish a *high-probability* PAC-Bayes **value-error** bound for a policy operating in a Markov decision process (MDP) when the training data are *dependent* trajectories—possibly gathered under an off-policy algorithm. In this section, we begin by fixing notation, then present the main results; all proofs are defered to Appendix B.

Let $\mathcal{M} = (\mathcal{S}, \mathcal{A}, \mathbb{P}, R, \gamma)$ be a discounted Markov Decision Process (MDP), where $\mathcal{S}$ and $\mathcal{A}$ are the state and action spaces, $\mathbb{P}$ is the transition kernel, $R$ is the reward function such that $R_t \in [0, R_{\max}]$, and $\gamma \in (0, 1)$ is the discount factor. A policy $\pi_\theta$ induces a (not necessarily *time-homogeneous*) Markov chain $\xi = (S_1, A_1, R_1, S_2, \ldots, S_H) \sim \nu, \mathbb{P}, \pi_\theta, R$, where $\nu$ denotes the initial state distribution and $H \leq \infty$ is the trajectory horizon (finite or infinite). Our analysis naturally extends to the infinite-horizon case ($H = \infty$).

We assume access to a dataset $\mathfrak{D} = \{\xi^{(1)}, \ldots, \xi^{(T)}\}$ of $T$ trajectories (*i.e.,* $N = HT$ transitions in total), collected using a behavior policy $\pi_\theta$, parameterized by $\theta \in \Theta$. The parameters $\theta$ are drawn from a distribution $\rho \in \mathcal{P}(\Theta)$, where $\Pi = \{\pi_\theta : \theta \in \Theta\}$ denotes the policy class. Henceforth, we write $\xi \sim \mathcal{M}$ (**resp.** $\mathfrak{D} \sim \mathcal{M}^{(T)}$) to denote sampling a trajectory (**resp.** a set $\mathfrak{D}$ of $T$ trajectories) under the environment dynamics $\mathbb{P}$, initial state distribution $\nu$, policy $\pi_\theta$, and reward function $R$, in order to avoid notational overload.

We define the discounted return of a trajectory and its expected value under policy $\pi_\theta$ as:

$$G(\xi) = \sum_{k=0}^{H-1} \gamma^k R_{k+1} \quad \text{and} \quad V_{\pi_\theta} = \mathbb{E}_{\xi \sim \mathcal{M}}[G(\xi)] \tag{4}$$

We now define the expected (true) loss and its empirical counterpart:

$$\mathcal{L}(\theta) = -\mathbb{E}_{\xi \sim \mathcal{M}}[G(\xi)] = \mathbb{E}_{\mathfrak{D} \sim \mathcal{M}^{(T)}}[\hat{\mathcal{L}}_{\mathfrak{D}}(\theta)], \quad \text{where} \quad \hat{\mathcal{L}}_{\mathfrak{D}}(\theta) = -\frac{1}{T}\sum_{j=1}^{T} G(\xi^{(j)}). \tag{5}$$

Following the PAC-Bayesian paradigm, we endow the parameter space $\Theta$ with a prior distribution $\mu \in \mathcal{P}(\Theta)$ selected independently of data and a posterior $\rho \in \mathcal{P}(\Theta)$ chosen after observing $\mathfrak{D}$. This formalism enables reasoning about randomized policies drawn from $\rho$ with guarantees based on their divergence from $\mu$. Crucially for our analysis, changing one transition in the data results in a quantifiable bounded effect on the empirical loss defined in equation 5:

**Lemma 3.1** (Bounded differences). *Let $\mathfrak{D}$ be a set of trajectories and $\theta \in \Theta$ be fixed policy parameters. Suppose we form $\bar{\mathfrak{D}}$ by changing one transition, say the transition at time step $h \in [H]$ of trajectory $j \in [T]$, where $\xi_h^{(j)} = (s, a, r, s')$ is replaced with $\bar{\xi}_h^{(j)} = (\bar{s}, \bar{a}, \bar{r}, \bar{s}')$. Then, there exists $c \in \mathrm{IR}_+^{H \times T}$ such that*

$$\left|\hat{\mathcal{L}}_{\mathfrak{D}}(\theta) - \hat{\mathcal{L}}_{\bar{\mathfrak{D}}}(\theta)\right| \leq \sum_{h'=1}^{H}\sum_{j'=1}^{T} c_{(h',j')} \, \mathbb{I}\left[\xi_{h'}^{(j')} = \bar{\xi}_{h'}^{(j')}\right] \tag{6}$$

Intuitively, $c_{(h,j)}$ quantifies the *transition-level influence* of altering the $(h, j)$-th state–action–reward tuple on the average return. A complete derivation—including a justification of why this bound covers propagation of the perturbed transition to future steps—is given in Appendix B.2. The result yields the explicit vector $c$ used in the main Theorem 3.2.

$$c_{(h,t)} = \frac{\gamma^{h-1} R_{\max}}{T}, \qquad \|c\|^2 = \frac{R_{\max}^2}{T(1-\gamma^2)}\left(1 - \gamma^{2H}\right). \tag{7}$$

## 3.2 MAIN RESULT

The bounded-differences property defined above is precisely what allows us to apply concentration inequalities for dependent data. Specifically, we leverage Paulin (2018)'s extension of McDiarmid's inequality to Markov chains, which provides concentration for functions satisfying bounded differences on Markovian sequences. The key insight is that while the transitions are temporally dependent, the bounded-differences condition with explicit constants $\|c\|^2$ enables us to control how perturbations propagate through the dependency structure.

Applying Paulin (2018)'s concentration result yields a tail bound on the deviation $\mathcal{L}(\theta) - \hat{\mathcal{L}}_{\mathfrak{D}}(\theta)$ (see equation 15) that depends explicitly on the mixing time $\tau_{\min}$ of the policy-induced Markov chain. $\tau_{\min}$ is the smallest number of steps after which the distribution of the chain's state is, in a statistical sense, nearly indistinguishable from its long-run or stationary distribution in Total Variation distance, no matter where the chain started. In other words, it measures how quickly the chain "forgets" its initial state and becomes well mixed. Combining the tail bound from equation 15 with the standard PAC-Bayesian change-of-measure technique gives our main result:

**Theorem 3.2.** *Let the reward function be bounded in $[0, R_{max}]$ and let $\mathcal{M}$ be a (not necessarily time-homogeneous) Markov Decision Process (MDP) induced by any policy $\pi_\theta$ such that it satisfies $\tau_{\min} < \infty$. For any prior $\mu$ over $\Pi$, any posterior $\rho$ chosen after interacting with the environment, and any $\delta \in (0,1)$, with probability at least $1 - \delta$ over the sample $\mathfrak{D}$ of $T$ trajectories with time horizon $H$:*

$$\mathbb{E}_{\theta \sim \rho}\left[\mathcal{L}(\theta) - \hat{\mathcal{L}}_{\mathfrak{D}}(\theta)\right] \leq \sqrt{\frac{R_{\max}^2 \tau_{\min}\left(1 - \gamma^{2H}\right)}{2T(1 - \gamma^2)}\left(\mathrm{KL}(\rho\|\mu) + \ln\frac{2}{\delta}\right)}. \tag{8}$$

The bound in 3.2 can be straightforwardly converted to a PAC-Bayes lower bound on the true expected value function $\mathbb{E}_{\theta \sim \rho}[V_{\pi_\theta}]$ (see Appendix B, equation 32), using the fact that $\mathcal{L}(\theta) = -V_{\pi_\theta}$ (equation 5) and a simple rearrangement of terms. The true expected value is lower-bounded by an empirical estimate minus an uncertainty term that accounts for limited data ($1/T$), temporal correlations ($\tau_{\min}$), and posterior complexity ($\mathrm{KL}(\rho\|\mu)$). This interpretation suggests a natural approach to policy optimization: select the posterior $\rho$ that maximizes this lower bound (equivalently, minimizes the upper bound in equation 8). Such a strategy would automatically balance exploitation (maximizing the empirical value) and theoretically-justified exploration (accounting for uncertainty).

## 3.3 KEY IMPROVEMENTS AND DISCUSSION

**Improved Scaling.** Previous PAC-Bayesian bounds for RL suffer from poor scaling with the discount factor. Fard et al. (2012) requires $N > R^4/(1 - \gamma)^4$ total time steps for non-vacuity, while Tasdighi et al. (2025) faces similar constraints. For $\gamma = 0.99$, this scaling renders such bounds practically vacuous. Our transition-level analysis achieves substantially better scaling: $T > \frac{R_{\max}^2 \tau_{\min}(1-\gamma^{2H})}{2(1-\gamma^2)}$ trajectories suffice. Although this may look prohibitive due to dependence on the number of trajectories, the key improvements are: **(i)** $\gamma$ appears with power 2 rather than in $(1 - \gamma)^4$, and **(ii)** collecting many short trajectories makes the numerator small since $(1 - \gamma^{2H}) < 1$.

**Explicit Mixing Time Dependence.** Unlike bounds for general mixing processes that depend on abstract coefficients, our result features explicit dependence on $\tau_{\min}$, the mixing time of the policy-induced Markov chain. This quantity is well studied and has a clear interpretation in terms of environment dynamics. While extending concentration inequalities from independent to mixing settings is sometimes viewed as straightforward, the reality involves several subtle challenges. Although the distribution of $X_t$ becomes close to the stationary distribution $\pi$ after $\tau_{\min}$ steps, this does not guarantee that $X_t$ is approximately independent of $X_0$, let alone that consecutive states $X_t$ and $X_{t+1}$ are independent. The dependence between observations decays exponentially with the mixing time, but achieving approximate independence typically requires waiting several multiples of $\tau_{\min}$, not just $\tau_{\min}$ itself.

Moreover, applying McDiarmid-type inequalities to RL requires establishing that the negative empirical return satisfies a bounded-differences condition with explicit, tractable constants (Lemma

3.1). This necessitates careful analysis of how perturbations at individual transitions propagate through the sequential structure to affect future states and rewards. Our transition-level analysis directly addresses this challenge by quantifying the error propagation through the Markov dependency structure. The proof is provided in Appendix B.4.

**Practical Tractability and Robustness.** The bound requires estimating $\tau_{\min}$, which presents both computational and robustness considerations. We estimate mixing time using autocorrelation decay of the reward signal, as this can be computed from streaming trajectories without storing full visitation counts. Alternative approaches using spectral methods could provide better estimates leading to tighter bounds but require sufficient state-action coverage to construct reliable empirical transition matrices which is infeasible when the state space is continuous.

Our estimation approach is robust to errors in a specific direction: if we overestimate $\tau_{\min}$, our bound remains valid but becomes looser, which is harmless in practice. However, underestimation can be problematic as it leads to overconfidence. To mitigate this, one can compute autocorrelation from multiple sources (*e.g.*, state features, value estimates) to cross-validate mixing time estimates. In practice, we err on the side of caution by using a conservative initial estimate and taking the maximum with the latest autocorrelation estimation, trading some tightness for reliability.

# 4 PB-SAC: A Practical Algorithm for PAC-Bayesian Reinforcement Learning

Translating our theoretical PAC-Bayesian bound into a practical learning algorithm requires addressing challenges of maintaining posterior distributions over policy parameters in deep RL. Our algorithm, **PAC-Bayes Soft Actor-Critic (PB-SAC)**, builds upon SAC while integrating PAC-Bayesian bounds and posterior-guided exploration (pseudo-code and illustrative figure in Appendix D).

PB-SAC's central insight is that policy parameters always represent the posterior mean, updated through standard SAC gradients during regular training, ensuring most learning follows proven SAC dynamics while periodic PAC-Bayesian updates refine the posterior and guide exploration. Following Zhang et al. (2024), we maintain a diagonal Gaussian posterior $\rho(\theta) = \mathcal{N}(\upsilon, \operatorname{diag}(\sigma^2))$ over flattened policy parameters with learnable mean $\upsilon$ and standard deviation $\sigma$. The prior $\mu$ undergoes periodic moving average updates toward the current posterior with linear decay, preventing KL divergence explosion while preserving bound validity and maintaining exploration capability as the prior stabilizes during training.

## 4.1 Posterior-Guided Exploration

During exploration, PB-SAC leverages the posterior distribution to implement uncertainty-driven exploration. Rather than standard $\epsilon$-greedy exploration, the algorithm samples policies from the posterior and selects actions that maximize Q-values under posterior uncertainty: $\theta_{explore} \leftarrow \arg\max_{\theta \sim \rho} Q(s, \pi_\theta(s))$. This posterior-guided exploration naturally balances exploitation of the mean policy (the current actor) when it yields the highest value, with exploration of alternative policies in regions of high posterior uncertainty where potentially superior policies may exist. This approach provides theoretical grounding for the exploration strategy through the PAC-Bayesian framework, ensuring that exploration is guided by uncertainty quantification rather than arbitrary randomness.

## 4.2 Alternating Optimization via PAC-Bayes-$\kappa$

The core challenge in optimizing our PAC-Bayesian bound lies in its structure: while the KL divergence term $\operatorname{KL}(\rho\|\mu)$ is convex in its first argument (The posterior parameters), the square root composition in our bound is not guaranteed to be convex, potentially leading to optimization difficulties. To address this challenge, we leverage an intermediate step (equation 23) in our proof detailed in Appendix B.5.4. This PAC-Bayes-$\kappa$ has a structure remarkably similar to the PAC-Bayes-$\lambda$ bound of Thiemann et al. (2017), derived from the classical PAC-Bayes-$kl$ bound for majority vote learning (see, *e.g.*, Germain et al. (2015), Corollary 21; Seeger (2003); Langford (2005)). Both

approaches share the key insight of introducing an auxiliary trade-off parameter—$\lambda$ in their case, $\kappa$ in ours—that transforms non-convex objectives into quasi-convex relaxations amenable to stable alternating optimization.

Our implementation optimizes the following PAC-Bayes-$\kappa$ objective to obtain the posterior $\rho$ then substitutes this optimized posterior into equation 8. As shown in our proof B.5.5, the main bound achieves its minimum over $\kappa^*$, yielding a tighter certificate:

$$\mathcal{L}(\rho, \kappa) = \mathbb{E}_{\theta \sim \rho} \left[ \hat{\mathcal{L}}_{\mathfrak{D}}(\theta) \right] + \frac{\mathrm{KL}(\rho \| \mu)}{\kappa} + \frac{\kappa \| c \|^2 \tau_{\min}}{8} \tag{9}$$

This decomposition enables stable alternating optimization: we optimize posterior parameters for fixed $\kappa$, then optimize $\kappa$ for fixed posterior parameters, ensuring convergence while maintaining the theoretical guarantees of our PAC-Bayesian bound.

### 4.3 POLICY-LEVEL REINFORCE TRICK

Even with the above decomposition, we cannot straightforwardly compute the gradient $\nabla_{(v,\sigma)} \mathbb{E}_{\theta \sim \rho} \left[ \hat{\mathcal{L}}_{\mathfrak{D}}(\theta) \right]$ because sampling cannot occur inside the gradient operation. To address this challenge, we employ a two-stage approach. First, we collect fresh rollouts during the PAC-Bayes update cycle using the mean policy, then sample policies from $\rho$ and evaluate their discounted returns on these rollouts using importance sampling. This ensures our bound computation accounts for distributional shift between the data-generating policy and the current posterior distribution. With estimated returns for each sampled policy, we apply the log-likelihood trick (REINFORCE) (Williams, 1992) at the policy level rather than the traditional action level. We prove this extension in Appendix C. This technique allows us to exchange gradient and expectation operations: $\mathbb{E}_{\theta \sim \rho} \left[ \nabla_{(v,\sigma)} \log \mathbb{P}_{v,\sigma}(\theta) \hat{\mathcal{L}}_{\mathfrak{D}}(\theta) \right]$, yielding a tractable sampling-based gradient estimator.

### 4.4 THE ADAPTIVE SAMPLING CURRICULUM

Our most critical innovation addresses the "posterior syncing shock" problem: when the posterior mean shifts significantly after PAC-Bayesian updates, critics become misaligned with the new policy distribution, creating a destabilizing feedback loop where inaccurate value estimates generate misleading gradients that further destabilize training through exploding gradients and learning instability.

PB-SAC resolves this through an adaptive sampling curriculum: immediately following PAC-Bayesian updates, we freeze the actor and employ high-rate posterior sampling (512 samples) to expose critics to the full posterior distribution for recalibration, then resume efficient learning with minimal sampling (1 sample, the posterior mean), achieving computational efficiency without sacrificing stability.

## 5 EXPERIMENTS

We now turn to empirical validation across representative continuous control tasks to demonstrate that our PAC-Bayesian framework delivers on this promise while maintaining competitive learning performance.

### 5.1 EXPERIMENTAL DESIGN

We evaluate PB-SAC on four MuJoCo continuous control environments (Towers et al., 2024; Tassa et al., 2018) spanning different complexity levels: HalfCheetah, Ant, Hopper, and Walker2d (in Appendix E). Our evaluation tracks two critical metrics: PAC-Bayesian certificate evolution (Figure 1, right) and learning performance relative to baselines (Figure 1, left)[1].

---

[1]The codebase is available in supplementary materials.

We run PB-SAC with PAC-Bayesian updates every 20,000 steps, employing our adaptive sampling curriculum (512 posterior samples during critic adaptation, posterior mean alone during regular training). We compare against vanilla SAC using identical network architectures and include PBAC Tasdighi et al. (2025), a PAC-Bayes deep exploration method, despite its primary strengths manifesting in sparse reward settings. Table 1 summarizes all hyperparameters.

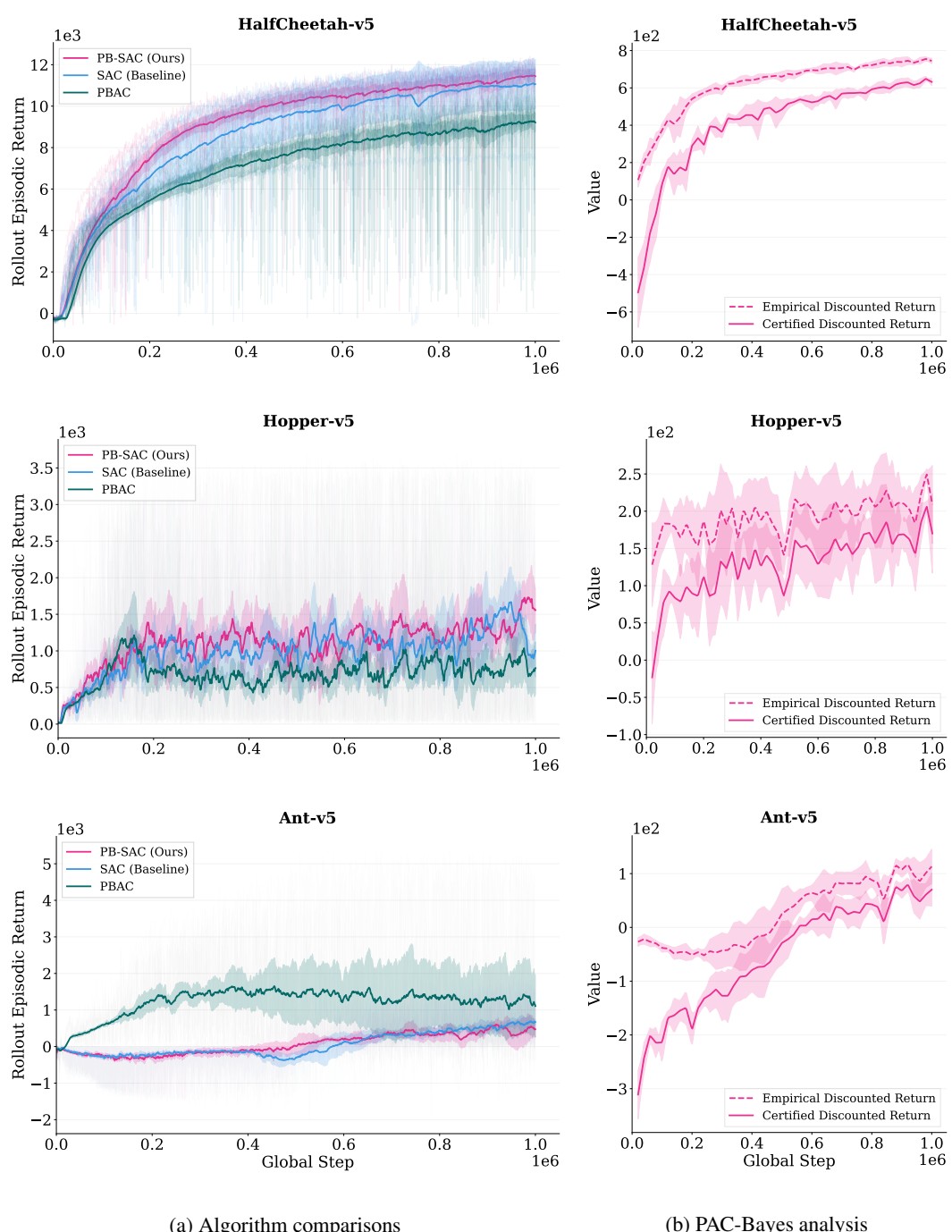

(a) Algorithm comparisons  (b) PAC-Bayes analysis

Figure 1: **(a)** Performance comparison between our ■ PB-SAC, its baseline ■ SAC, and ■ PBAC from Tasdighi et al. (2025); **(b)** PAC-Bayes analysis of PB-SAC across environments. The empirical discounted return (dashed line) corresponds to $\mathbb{E}_{\theta \sim \rho}[-\hat{\mathcal{L}}_{\mathfrak{D}}(\theta)]$, and the certified discounted return (solid line) corresponds to the lower bound on $\mathbb{E}_{\theta \sim \rho}[-\mathcal{L}(\theta)]$ provided by Theorem 3.2 (after rearranging the terms).

## 5.2 THE TIGHTENING OF PERFORMANCE CERTIFICATES

Figure 1b reveals the most compelling aspect of our results: the PAC-Bayesian bounds consistently tighten throughout training, tracking the improvement in learned policies. In HalfCheetah, the bounds become informative within $100k$ steps and continue tightening as performance improves, demonstrating that our certificates provide genuine confidence estimates rather than vacuous guarantees, without hindering learning. Hopper exhibits similar bound evolution, though with slower convergence reflecting its increased complexity. Particularly noteworthy is the bounds' behavior during performance fluctuations: they appropriately widen during periods of high variance while tightening when performance stabilizes (e.g., HalfCheetah between $100k$ and $200k$ steps). This stability stems from our decaying moving average prior updates, which prevent KL explosion in early stages; without this mechanism, KL explosion hinders learning by pulling the posterior back toward the prior rather than allowing improvement.

Ant presents the most challenging test case due to its 3D dynamics and large state space. Even here, our bounds remain meaningful throughout training, tightening during improvement periods (within $500k$ steps) and providing conservative estimates during temporary performance drops (between $800k$ and $900k$ steps). The bounds capture the qualitative behavior predicted by our theory: environments with slower mixing yield more gradual certificate improvement.

## 5.3 EMPIRICAL VALIDATION ACROSS ENVIRONMENTS

The learning curves in Figure 1a address a fundamental concern about PAC-Bayesian RL: whether theoretical guarantees necessitate performance sacrifices. Our results demonstrate that PB-SAC matches or exceeds SAC's sample efficiency across all tested environments. In HalfCheetah and Hopper, PB-SAC shows superior performance compared to both SAC and PBAC. This validates our architectural choices, particularly the policy-posterior synchronization that ensures most learning follows proven SAC dynamics. The periodic PAC-Bayesian updates serve as refinements rather than disruptions, guided by the adaptive sampling curriculum that prevents critic destabilization.

In Ant, PBAC initially outperforms both methods thanks to its multi-objective design accounting for diversity, coherence, and propagation, beneficial when rewards are less informative. However, excessive exploration eventually backfires, causing performance degradation in later training. PB-SAC matches its baseline with slightly faster convergence around $500k$ steps, though lagging slightly at the end. While not designed specifically for deep exploration, it consistently matches baseline performance while providing theoretical guarantees.

# 6 CONCLUSION

We introduced PB-SAC, a PAC-Bayesian actor-critic algorithm that bridges the gap between learning theory and practical deep RL by making practical use of non-vacuous performance certificates as active components of the learning process. Our theoretical contribution (a PAC-Bayesian bound with explicit mixing-time dependence and improved discount factor scaling) enables meaningful generalization guarantees in sequential decision-making. Algorithmically, our key innovations include policy-posterior synchronization, adaptive sampling curriculum for critic adaptation, posterior-guided exploration, and prior moving average that maintains competitive performance while providing formal certificates.

Several limitations warrant consideration. Our approach requires accurate mixing time estimation (underestimation leads to overconfident bounds), faces computational overhead that may limit scalability to large networks, and uses KL divergence which, despite its analytical convenience, is unstable when distributions diverge significantly and does not respect the geometry of the parameter space (Viallard et al., 2023). Future work could explore efficient posterior approximations, meaningful representations (layer/parameter-level rather than full network posteriors), adaptive mixing time estimation, Wasserstein distance alternatives to KL (Amit et al., 2022; Haddouche & Guedj, 2023; Viallard et al., 2023), and extensions beyond actor-critic frameworks. Despite these constraints, our results establish PAC-Bayesian RL as viable for certified performance in modern reinforcement learning, particularly valuable in high-stakes applications requiring theoretical guarantees.

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

# A MATHEMATICAL TOOLS

**Lemma A.1** (Markov's Inequality). *For any random variable $X$ such that $\mathbb{E}[|X|] = \mu$, for any $a > 0$, we have*

$$\mathbb{P}\{|X| \geq a\} \leq \frac{\mu}{a}.$$

**Lemma A.2** (Change of measure). *For any measurable function $f : \Theta \to \mathbb{R}$ and distributions $\mu, \rho \in \mathcal{P}(\Theta)$:*

$$\mathbb{E}_{\theta \sim \rho}[f(\theta)] \leq \mathrm{KL}(\rho\|\mu) + \ln \mathbb{E}_{\theta \sim \mu}[\exp(f(\theta))] \tag{10}$$

*where $\mathrm{KL}(\rho\|\mu)$ is the Kullback-Leibler divergence between $\rho$ and $\mu$.*

## A.1 CONCENTRATION FOR MARKOV CHAINS VIA MARTON COUPLING

We use Paulin's extension of McDiarmid's bounded-difference inequality to Markov chains (Paulin, 2018). This extension provides concentration inequalities for functions of dependent random variables, with constants that depend on the mixing properties of the chain.

### A.1.1 MARTON COUPLING AND MIXING TIME

The key insight in Paulin's approach (Paulin, 2018) is to use a coupling structure known as Marton coupling, which quantifies the dependency between random variables in a Markov chain. For a Markov chain $X = (X_1, \ldots, X_N)$ on state space $\Lambda = \Lambda_1 \times \ldots \times \Lambda_N$, a Marton coupling provides a way to couple the distributions of future states conditioned on different past states.

Let $\tau(\varepsilon)$ denote the mixing time of the chain $X$ in total variation distance, defined as the minimal $t$ such that for every $1 \leq i \leq N - t$ and $x, y \in \Lambda_i$:

$$d_{TV}(\mathcal{L}(X_{i+t}|X_i = x), \mathcal{L}(X_{i+t}|X_i = y)) \leq \varepsilon \tag{11}$$

We define the normalized mixing time parameter $\tau_{\min}$ as:

$$\tau_{\min} = \inf_{0 \leq \varepsilon < 1} \tau(\varepsilon)\left(\frac{2-\varepsilon}{1-\varepsilon}\right)^2 \tag{12}$$

### A.1.2 MCDIARMID'S INEQUALITY FOR MARKOV CHAINS

For a function $f(X)$ satisfying the bounded-differences property: for any $x, y \in \Lambda$,

$$f(x) - f(y) \leq \sum_{i=1}^{N} c_i \mathbb{I}[x_i \neq y_i] \tag{13}$$

where $c \in \mathbb{R}_+^N$ and $\mathbb{I}[\text{condition}]$ is the indicator function, Paulin's theorem gives:

$$\Pr\big(|f(X) - \mathbb{E}f(X)| \geq t\big) \leq 2\exp\big(-2t^2/\|c\|^2 \tau_{\min}\big). \qquad (14)$$

The norm $\|c\|^2$ is defined as $\sum_{i=1}^N c_i^2$.

### A.1.3 APPLICATION TO BOUNDED DIFFERENCES IN MDPs

For Markov decision processes, this inequality is particularly useful when analyzing the difference between value functions. If perturbing a single transition can change the value by at most $c_i$, then the total effect on a function of trajectories is bounded by the above concentration inequality, with the mixing time of the MDP properly accounting for the propagation of the perturbation through future states.

## B DERIVATION OF PAC-BAYES VALUE-ERROR BOUND FOR RL

### B.1 BOUNDED-DIFFERENCES PROPERTY FOR MDP TRAJECTORIES

We begin by recalling the definitions of discounted return for a trajectory $\xi$ and the corresponding value function from Section 3:

$$G(\xi) = \sum_{k=0}^{H-1} \gamma^k R_{k+1}$$
$$V_{\pi_\theta} = \mathbb{E}_{\xi \sim \mathcal{M}}[G(\xi)]$$

As defined in equation 5, our empirical and expected losses are:

$$\hat{\mathcal{L}}_{\mathfrak{D}}(\theta) = -\frac{1}{T} \sum_{j=1}^T G(\xi^{(j)})$$
$$\mathcal{L}(\theta) = -\mathbb{E}_{\xi \sim \mathcal{M}}[G(\xi)] = -V_{\pi_\theta}$$

To apply McDiarmid's inequality for Markov chains, we must establish the bounded-differences condition for our empirical loss. Specifically, we need to show that replacing one transition in a trajectory affects $\hat{\mathcal{L}}_{\mathfrak{D}}(\theta)$ by at most $\sum_{h=1}^H \sum_{j=1}^T c_{(h,j)} \mathbb{I}[\xi_h^{(j)} \neq \bar{\xi}_h^{(j)}]$, where $c \in \mathbb{R}_+^{H \times T}$ and $\mathbb{I}$ is the indicator function.

### B.2 QUANTIFYING THE IMPACT OF PERTURBED TRANSITIONS

Suppose we replace a single transition at position $h$ in trajectory $j$. The change in the discounted return of this trajectory is bounded by:

$$|G(\xi^{(j)}) - G(\bar{\xi}^{(j)})| = |\gamma^{h-1}(R_h - \bar{R}_h) + \text{effects on future rewards}|$$
$$\leq \gamma^{h-1} R_{\max} + \text{effects on future rewards}$$

Crucially, this perturbation affects not only the immediate reward but potentially all subsequent transitions and rewards in that trajectory. The change in our empirical loss is therefore bounded by:

$$|\hat{\mathcal{L}}_{\mathfrak{D}}(\theta) - \hat{\mathcal{L}}_{\bar{\mathfrak{D}}}(\theta)| \leq \frac{\gamma^{h-1} R_{\max}}{T} = c_{(h,j)}$$

## B.3 DERIVATION OF $\|c\|^2$ FOR THE PAC-BAYES BOUND

To apply McDiarmid's inequality for Markov chains as developed by Paulin (2018), we need to compute $\|c\|^2$:

$$\|c\|^2 = \sum_{j=1}^{T} \sum_{h=1}^{H} c^2(h,j)$$

$$= \frac{R_{\max}^2}{T^2} \cdot T \sum_{h=1}^{H} \gamma^{2(h-1)}$$

$$= \frac{R_{\max}^2}{T} \underbrace{\sum_{h=0}^{H-1} \gamma^{2h}}_{\text{finite geometric series}}$$

$$= \frac{R_{\max}^2}{T} \cdot \frac{1-\gamma^{2H}}{1-\gamma^2}.$$

For infinite-horizon settings where $H \to \infty$ and $\gamma < 1$, the series converges to $1/(1-\gamma^2)$, this simplifies to

$$\|c\|^2 = \frac{R_{\max}^2}{T(1-\gamma^2)}.$$

## B.4 FULL ACCOUNTING OF PERTURBATION PROPAGATION EFFECTS

A critical question is whether our derivation of $\|c\|^2$ fully accounts for the propagation of perturbations through the trajectory. Since a perturbation at step $h$ in trajectory $j$ affects all subsequent transitions in that trajectory, the bounded-differences indicator is 1 for every $(h',j)$ with $h' \geq h$.

For a perturbation at step $h$ in trajectory $j$, the sum of corresponding coefficients is:

$$\sum_{h'=h}^{H} c_{(h',j)} = \frac{R_{\max}}{T} \sum_{k=0}^{H-h} \gamma^{h-1+k} = \frac{R_{\max}\gamma^{h-1}}{T} \cdot \frac{1-\gamma^{H-h+1}}{1-\gamma}$$

The actual maximum change in discounted return from this perturbation (worst case: reward changes from 0 to $R_{\max}$) is:

$$|G(\xi^{(j)}) - G(\bar{\xi}^{(j)})| \leq R_{\max}\gamma^{h-1} \sum_{k=0}^{H-h} \gamma^k = R_{\max}\gamma^{h-1} \cdot \frac{1-\gamma^{H-h+1}}{1-\gamma}$$

When divided by $T$ (because $\hat{\mathcal{L}}_{\mathfrak{D}}(\theta)$ averages over $T$ trajectories), we get exactly the same quantity as the sum of coefficients above. Therefore, the bounded-differences condition holds with equality, confirming that our derivation of $\|c\|^2$ fully accounts for all propagation effects without requiring additional constants.

This careful accounting of propagation effects allows us to apply McDiarmid's inequality for Markov chains to obtain the PAC-Bayes bound in Theorem 3.2 with the correct constants.

## B.5 DERIVATION OF THE PAC-BAYES BOUND

Having established the bounded-differences property and quantified the impact of perturbations via $\|c\|^2$, we now derive the PAC-Bayes bound on the expected difference between empirical and true losses.

### B.5.1 FROM MCDIARMID TO MOMENT GENERATING FUNCTION

McDiarmid's inequality for Markov chains (equation 14) provides a concentration inequality on the deviation between empirical and expected losses. From this, we can derive a bound on the moment generating function (MGF) as shown by Paulin (2018):

**Lemma B.1** (MGF bound for Markov chains). *For any $\kappa > 0$ and policy parameters $\theta \in \Theta$:*

$$\mathbb{E}_{\mathfrak{D} \sim \mathcal{M}^{(T)}} \left[ \exp \left( \kappa(\hat{\mathcal{L}}_{\mathfrak{D}}(\theta) - \mathcal{L}(\theta)) \right) \right] \leq \exp \left( \frac{\kappa^2 \|c\|^2 \tau_{\min}}{8} \right) \tag{15}$$

*where $\tau_{\min}$ is the mixing time of the Markov chain induced by policy $\pi_\theta$.*

### B.5.2 PAC-BAYES CHANGE OF MEASURE

Now we can follow the standard PAC-Bayes derivation. Let $\Theta$ be our parameter space and let $\mu \in \mathcal{P}(\Theta)$ be a prior distribution over $\Theta$ chosen independently of the data. For any posterior distribution $\rho \in \mathcal{P}(\Theta)$ (which may depend on $\mathfrak{D}$), we apply the change-of-measure inequality (Donsker–Varadhan (Donsker & Varadhan, 1983) variational formula)

Let $f(\theta) = \kappa(\hat{\mathcal{L}}_{\mathfrak{D}}(\theta) - \mathcal{L}(\theta))$. Applying Lemma A.2:

$$\mathbb{E}_{\theta \sim \rho}[\kappa(\hat{\mathcal{L}}_{\mathfrak{D}}(\theta) - \mathcal{L}(\theta))] \leq \text{KL}(\rho\|\mu) + \ln \mathbb{E}_{\theta \sim \mu}[\exp(\kappa(\hat{\mathcal{L}}_{\mathfrak{D}}(\theta) - \mathcal{L}(\theta)))] \tag{16}$$

### B.5.3 COMBINING WITH THE MGF BOUND

Taking the expectation with respect to $\mathfrak{D} \sim \mathcal{M}^{(T)}$ on both sides:

$$\mathbb{E}_{\mathfrak{D}} \mathbb{E}_{\theta \sim \rho}[\kappa(\hat{\mathcal{L}}_{\mathfrak{D}}(\theta) - \mathcal{L}(\theta))] \leq \text{KL}(\rho\|\mu) + \mathbb{E}_{\mathfrak{D}} \ln \mathbb{E}_{\theta \sim \mu}[\exp(\kappa(\hat{\mathcal{L}}_{\mathfrak{D}}(\theta) - \mathcal{L}(\theta)))] \tag{17}$$

By Jensen's inequality, since $\ln$ is concave:

$$\mathbb{E}_{\mathfrak{D}} \mathbb{E}_{\theta \sim \rho}[\kappa(\hat{\mathcal{L}}_{\mathfrak{D}}(\theta) - \mathcal{L}(\theta))] \leq \text{KL}(\rho\|\mu) + \ln \mathbb{E}_{\mathfrak{D}} \mathbb{E}_{\theta \sim \mu}[\exp(\kappa(\hat{\mathcal{L}}_{\mathfrak{D}}(\theta) - \mathcal{L}(\theta)))] \tag{18}$$

By Fubini's theorem (exchanging the order of expectations) and Lemma B.1:

$$\mathbb{E}_{\mathfrak{D}} \mathbb{E}_{\theta \sim \rho}[\kappa(\hat{\mathcal{L}}_{\mathfrak{D}}(\theta) - \mathcal{L}(\theta))] \leq \text{KL}(\rho\|\mu) + \ln \mathbb{E}_{\theta \sim \mu} \mathbb{E}_{\mathfrak{D}}[\exp(\kappa(\hat{\mathcal{L}}_{\mathfrak{D}}(\theta) - \mathcal{L}(\theta)))] \tag{19}$$

$$\leq \text{KL}(\rho\|\mu) + \ln \mathbb{E}_{\theta \sim \mu} \left[ \exp \left( \frac{\kappa^2 \|c\|^2 \tau_{\min}}{8} \right) \right] \tag{20}$$

$$= \text{KL}(\rho\|\mu) + \frac{\kappa^2 \|c\|^2 \tau_{\min}}{8} \tag{21}$$

Dividing by $\kappa > 0$:

$$\mathbb{E}_{\mathfrak{D}} \mathbb{E}_{\theta \sim \rho}[\hat{\mathcal{L}}_{\mathfrak{D}}(\theta) - \mathcal{L}(\theta)] \leq \frac{\text{KL}(\rho\|\mu)}{\kappa} + \frac{\kappa \|c\|^2 \tau_{\min}}{8} \tag{22}$$

### B.5.4 HIGH-PROBABILITY BOUND VIA MARKOV'S INEQUALITY

Now, we convert this expectation bound into a high-probability bound. By Markov's inequality A.1, for any non-negative random variable $X$ and $\delta > 0$:

With probability at least $1 - \delta$:

$$\mathbb{E}_{\theta \sim \rho}[\hat{\mathcal{L}}_{\mathfrak{D}}(\theta) - \mathcal{L}(\theta)] \leq \frac{\text{KL}(\rho\|\mu) + \ln \frac{2}{\delta}}{\kappa} + \frac{\kappa \|c\|^2 \tau_{\min}}{8} \tag{23}$$

### B.5.5 OPTIMIZING THE BOUND

To tighten the bound, we minimize the right-hand side with respect to $\kappa > 0$. Taking the derivative and setting it to zero:

$$\frac{\partial}{\partial \kappa} \left( \frac{\mathrm{KL}(\rho\|\mu) + \ln\frac{2}{\delta}}{\kappa} + \frac{\kappa\|c\|^2\tau_{\min}}{8} \right) = 0 \tag{24}$$

$$-\frac{\mathrm{KL}(\rho\|\mu) + \ln\frac{2}{\delta}}{\kappa^2} + \frac{\|c\|^2\tau_{\min}}{8} = 0 \tag{25}$$

Solving for the optimal $\kappa^*$:

$$\kappa^* = \sqrt{\frac{8(\mathrm{KL}(\rho\|\mu) + \ln\frac{2}{\delta})}{\|c\|^2\tau_{\min}}} \tag{26}$$

Substituting $\kappa^*$ back into our bound:

$$\mathbb{E}_{\theta\sim\rho}[\hat{\mathcal{L}}_{\mathfrak{D}}(\theta) - \mathcal{L}(\theta)] \leq \frac{\mathrm{KL}(\rho\|\mu) + \ln\frac{2}{\delta}}{\kappa^*} + \frac{\kappa^*\|c\|^2\tau_{\min}}{8} \tag{27}$$

$$= \sqrt{\frac{\|c\|^2\tau_{\min}(\mathrm{KL}(\rho\|\mu) + \ln\frac{2}{\delta})}{8}} + \sqrt{\frac{\|c\|^2\tau_{\min}(\mathrm{KL}(\rho\|\mu) + \ln\frac{2}{\delta})}{8}} \tag{28}$$

$$= \sqrt{\frac{\|c\|^2\tau_{\min}(\mathrm{KL}(\rho\|\mu) + \ln\frac{2}{\delta})}{2}} \tag{29}$$

### B.5.6 FINAL BOUND

Finally, substituting the expression for $\|c\|^2$ from equation 7:

$$\mathbb{E}_{\theta\sim\rho}[\hat{\mathcal{L}}_{\mathfrak{D}}(\theta) - \mathcal{L}(\theta)] \leq \sqrt{\frac{\frac{R_{\max}^2}{T} \cdot \frac{1-\gamma^{2H}}{1-\gamma^2} \cdot \tau_{\min} \cdot (\mathrm{KL}(\rho\|\mu) + \ln\frac{2}{\delta})}{2}} \tag{30}$$

$$= \sqrt{\frac{R_{\max}^2\tau_{\min}(1-\gamma^{2H})}{2T(1-\gamma^2)}\left(\mathrm{KL}(\rho\|\mu) + \ln\frac{2}{\delta}\right)} \tag{31}$$

Recalling that $\mathcal{L}(\theta) = -V_{\pi_\theta}$ from equation 5, we obtain the PAC-Bayes lower bound on the expected value function:

$$\mathbb{E}_{\theta\sim\rho}[V_{\pi_\theta}] \geq \mathbb{E}_{\theta\sim\rho}[-\hat{\mathcal{L}}_{\mathfrak{D}}(\theta)] - \sqrt{\frac{R_{\max}^2\tau_{\min}(1-\gamma^{2H})}{2T(1-\gamma^2)}\left(\mathrm{KL}(\rho\|\mu) + \ln\frac{2}{\delta}\right)} \tag{32}$$

where The true expected value is lower-bounded by an empirical estimate minus an uncertainty term that accounts for limited data ($1/T$), temporal correlations ($\tau_{\min}$), and posterior complexity ($\mathrm{KL}(\rho\|\mu)$).

### B.6 A NOTE ON THE MARKOV ASSUMPTION FOR BELLMAN ERRORS

The work of Tasdighi et al. (2025) makes an interesting theoretical contribution by modeling the sequence of Bellman errors as a Markov chain. While this approach provides valuable insights, it is worth examining the conditions under which this assumption holds.

We present a simple illustrative example that highlights when the Markov property may not apply to Bellman error sequences. Consider a basic MDP with four states $\{A, B, C, D\}$ and the following transition dynamics with discount factor $\gamma = 0$:

- State $A$ transitions to state $C$ with reward $r = 0$
- State $B$ transitions to state $D$ with reward $r = 0$
- State $C$ has a self-loop with reward $r = +1$
- State $D$ has a self-loop with reward $r = -1$

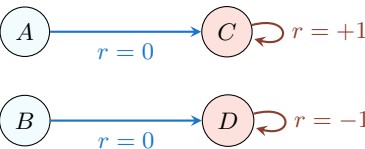

Figure 2: A basic four state MDP

Using a value function $V \equiv 0$ that assigns zero value to all states, we can compute the Bellman errors:

$$\delta_t(A) = r(A) + \gamma \max_a \mathbb{E}[V(s')|s = A, a] - V(A) = 0 + 0 \cdot V(C) - 0 = 0 \tag{33}$$

$$\delta_t(B) = r(B) + \gamma \max_a \mathbb{E}[V(s')|s = B, a] - V(B) = 0 + 0 \cdot V(D) - 0 = 0 \tag{34}$$

Both states $A$ and $B$ yield the same Bellman error $\delta_t = 0$ at time $t$. However, the subsequent errors differ:

$$\delta_{t+1}(C) = r(C) + \gamma \max_a \mathbb{E}[V(s')|s = C, a] - V(C) = +1 + 0 \cdot V(C) - 0 = +1 \tag{35}$$

$$\delta_{t+1}(D) = r(D) + \gamma \max_a \mathbb{E}[V(s')|s = D, a] - V(D) = -1 + 0 \cdot V(D) - 0 = -1 \tag{36}$$

This example demonstrates that the current Bellman error value $\delta_t = 0$ alone does not uniquely determine the distribution of $\delta_{t+1}$, which depends on the underlying state that generated the current error.

This observation suggests that the Markov assumption for Bellman errors may require additional conditions or refinements to hold more generally. Such considerations could be valuable for future theoretical developments building upon the framework proposed by Tasdighi et al. (2025).

## C REINFORCE TRICK FOR POLICY-LEVEL GRADIENTS

We prove that the REINFORCE trick can be extended from action-level to policy-level gradients, enabling tractable optimization of expectations over policy parameters.

**Theorem C.1** (Policy-Level REINFORCE). *For a posterior distribution $\rho = \mathcal{N}(v, \mathrm{diag}(\sigma^2))$ over policy parameters $\theta$, the gradient of the expected return can be computed as:*

$$\nabla_{v,\sigma} \mathbb{E}_{\theta \sim \rho}\left[\mathbb{E}[R_\tau|\pi_\theta]\right] = \mathbb{E}_{\theta \sim \rho}\left[\nabla_{v,\sigma} \log \mathbb{P}_{v,\sigma}(\theta) \cdot \mathbb{E}[R_\tau|\pi_\theta]\right] \tag{37}$$

*Proof.* Starting with the gradient of the expected return of a policy over the posterior distribution:

$$\nabla_{v,\sigma} \mathbb{E}_{\theta \sim \rho}\left[\mathbb{E}[R_\tau|\pi_\theta]\right] = \nabla_{v,\sigma} \sum_{\theta_i} \mathbb{P}_{v,\sigma}(\theta_i) \sum_a \pi_{\theta_i}(a) \underbrace{\mathbb{E}[R_\tau|A_\tau = a]}_{q(a)} \tag{38}$$

Moving the gradient inside the summation:

$$= \sum_{\theta_i} \nabla_{v,\sigma} \mathbb{P}_{v,\sigma}(\theta_i) \sum_a \pi_{\theta_i}(a) q(a) \tag{39}$$

where $q(a) = \mathbb{E}[R_\tau | A_\tau = a]$ is the state-action value for notational convenience.

Multiplying by $\frac{\mathbb{P}_{v,\sigma}(\theta_i)}{\mathbb{P}_{v,\sigma}(\theta_i)}$:

$$= \sum_{\theta_i} \left[ \left( \sum_a \pi_{\theta_i}(a) q(a) \right) \frac{\mathbb{P}_{v,\sigma}(\theta_i)}{\mathbb{P}_{v,\sigma}(\theta_i)} \nabla_{v,\sigma} \mathbb{P}_{v,\sigma}(\theta_i) \right] \tag{40}$$

Applying the log-derivative trick $\nabla_{v,\sigma} \mathbb{P}_{v,\sigma}(\theta_i) = \mathbb{P}_{v,\sigma}(\theta_i) \nabla_{v,\sigma} \log \mathbb{P}_{v,\sigma}(\theta_i)$:

$$= \sum_{\theta_i} \left[ \mathbb{P}_{v,\sigma}(\theta_i) \left( \sum_a \pi_{\theta_i}(a) q(a) \right) \nabla_{v,\sigma} \log \mathbb{P}_{v,\sigma}(\theta_i) \right] \tag{41}$$

Simplifying and recognizing that $\sum_a \pi_{\theta_i}(a) q(a) = \mathbb{E}[R_\tau | \pi_{\theta_i}]$:

$$= \sum_{\theta_i} \mathbb{P}_{v,\sigma}(\theta_i) \left( \mathbb{E}[R_\tau | \pi_{\theta_i}] \right) \nabla_{v,\sigma} \log \mathbb{P}_{v,\sigma}(\theta_i) \tag{42}$$

Converting back to expectation form:

$$= \mathbb{E}_{\theta \sim \rho} \left[ \nabla_{v,\sigma} \log \mathbb{P}_{v,\sigma}(\theta) \cdot \mathbb{E}[R_\tau | \pi_\theta] \right] \tag{43}$$

$\blacksquare$

This result enables us to estimate the gradient via sampling: we sample policies $\{\theta_i\}$ from the posterior $\rho$, evaluate their expected returns, and compute the weighted gradient of the log-probability density. This extends the classical REINFORCE trick from action spaces to parameter spaces, allowing efficient optimization of posterior distributions over policies.

## D PB-SAC ALGORITHM

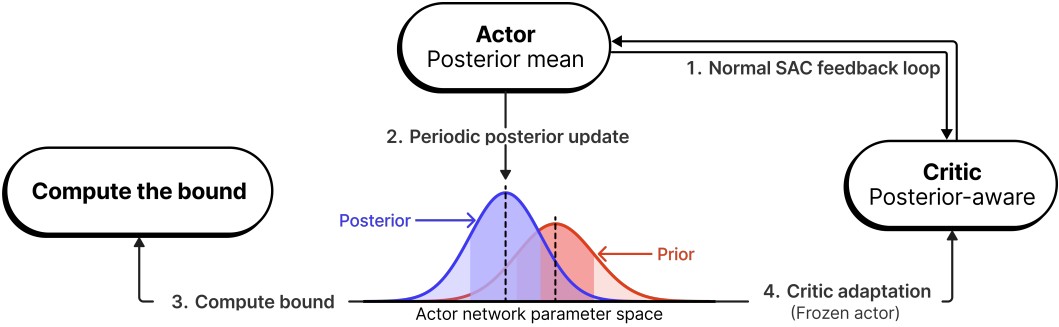

Figure 3: Illustration of our algorithm PB-SAC

## D.1 PSEUDO CODE

---

**Algorithm 1:** PAC-Bayes Soft Actor-Critic (PB-SAC)

---

**Result:** Policy $\pi_\theta$, posterior $\rho$, PAC-Bayes bound
**Init:** Actor $\pi_\theta$, Critics $Q_1, Q_2$, replay buffer $\mathfrak{D}$
**Init:** Posterior $\rho(\theta) = \mathcal{N}(\upsilon, \text{diag}(\sigma^2))$ where $\upsilon$ are the initial actor parameters
**Init:** Prior $\mu(\theta) = \mathcal{N}(\upsilon, \text{diag}(\sigma^2))$
**Init:** actor_frozen = False, prior moving average decay $\iota = 0.99$

```
1  for t = 1, 2, ... do
2      if not actor_frozen then
3          /* Standard SAC Training + Posterior-Guided Exploration  */
4          if random() < ε_explore then
               /* Select policy maximizing Q-value from posterior      */
5              θ_explore ← arg max_{θ∼ρ} Q(s, π_θ(s))
6              a ← π_{θ_explore}(s)
7          else
8              a ← π_θ(s)                        // Current policy (posterior mean)
9          end
10         Execute action a and store transition in 𝔇
11         if |𝔇| ≥ batch_size then
12             Update critics Q_1, Q_2 with standard SAC loss
13             Update actor π_θ with standard SAC loss
14             υ ← θ                    // Sync posterior mean to current actor
15         end
16     else
17         /* Critic Adaptation Phase (post PAC-Bayes update)         */
18         Sample multiple policies {θ_i} ∼ ρ with high sampling rate
19         Compute critic targets averaged over policy samples {θ_i}
20         Update critics Q_1, Q_2 using averaged targets
21         if adaptation steps completed then
22             actor_frozen ← False
23         end
24     end
25     /* PAC-Bayes Update Cycle                                      */
26     if t mod pb_update_freq = 0 then
27         𝔇_rollouts ← collect_fresh_rollouts()          // With the current policy
28         τ_min ← estimate_mixing_time(𝔇_rollouts)
29         𝔇_train, 𝔇_test ← split(𝔇_rollouts)
30         Compute discounted returns G_IS with importance sampling on 𝔇_train
           /* Alternating optimization                                */
31         for epoch = 1, ..., pb_epochs do
               /* Optimize posterior parameters for fixed κ           */
32             σ, υ ← arg min_{(σ,υ)} ℒ(ρ, κ)                          // equation 9
               /* Optimize κ for fixed posterior                      */
33             κ ← arg min_{κ'} ℒ(ρ, κ')                               // equation 9
34         end
35         bound ← compute_pac_bayes_bound(𝔇_test, KL(ρ‖μ), τ_min)
36         load_policy_params(υ)                  // Sync actor to posterior mean
37         actor_frozen ← True                    // Initiate critic adaptation
38     end
39     /* Prior Reset for Maintained Exploration                      */
40     if t mod pb_reset_freq = 0 then
41         μ ← ι · ρ + (1 − ι) · μ                 // Moving average prior update
42         Linearly decay ι
43     end
44 end
45 return π_θ, ρ, bound
```

# E MORE RESULTS

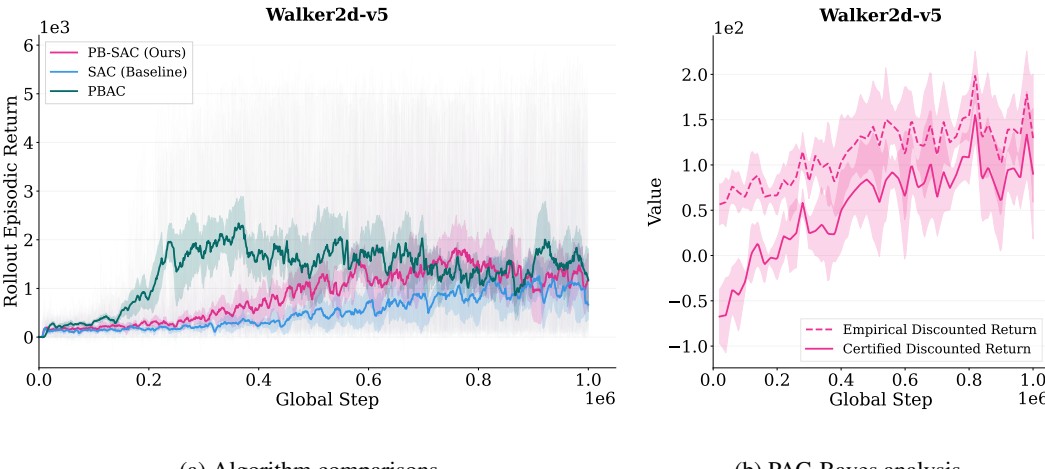

(a) Algorithm comparisons        (b) PAC-Bayes analysis

Figure 4: **(a)** Performance comparison between our ■ PB-SAC, its baseline ■ SAC, and ■ PBAC from Tasdighi et al. (2025); **(b)** PAC-Bayes analysis of PB-SAC across environments. The empirical discounted return (dashed line) corresponds to $\mathbb{E}_{\theta \sim \rho}[-\hat{\mathcal{L}}_{\mathfrak{D}}(\theta)]$, and the certified discounted return (solid line) corresponds to the lower bound on $\mathbb{E}_{\theta \sim \rho}[-\mathcal{L}(\theta)]$ provided by Theorem 3.2 (after rearranging the terms).

# F HYPERPARAMETER SELECTION

We carefully selected hyperparameters for our PAC-Bayes Soft Actor-Critic (PB-SAC) implementation to balance performance, sample efficiency, and theoretical guarantees. the common parameters with SAC are left unchanged, while we take the original hyperparameters of PBAC from the paper Tasdighi et al. (2025). Table 1 summarizes it all.

Table 1: Hyperparameter Comparison for MuJoCo Continuous Control Tasks

| Hyperparameter | PB-SAC (Our Algorithm) | SAC (Baseline) | PBAC |
|---|---|---|---|
| **Common SAC Parameters** | | | |
| Total Timesteps | $1 \times 10^6$ | $1 \times 10^6$ | $1 \times 10^6$ |
| Discount Factor ($\gamma$) | 0.99 | 0.99 | 0.99 |
| Soft Update Coefficient ($\tau$) | 0.005 | 0.005 | 0.005 |
| Batch Size | 256 | 256 | 256 |
| Replay Buffer Size | $1 \times 10^6$ | $1 \times 10^6$ | $1 \times 10^6$ |
| Initial Temperature ($\alpha$) | 0.2 | 0.2 | 0.2 |
| Temperature Learning Rate | $3 \times 10^{-4}$ | $3 \times 10^{-4}$ | $3 \times 10^{-4}$ |
| Target Update Frequency | 1 | 1 | 1 |
| **Algorithm-Specific Parameters** | | | |
| Actor Learning Rate | $3 \times 10^{-4}$ | $3 \times 10^{-4}$ | $3 \times 10^{-4}$ |
| Critic Learning Rate | $1 \times 10^{-3}$ | $1 \times 10^{-3}$ | $3 \times 10^{-4}$ |
| Learning Starts | 5,000 | 5,000 | 10,000 |
| Training Frequency | 2 | 2 | 1 |
| Automatic $\alpha$ Tuning | ✓ | ✓ | × |
| Multi-Head Architecture | × | × | ✓ |
| Ensemble of Critics | × | × | ✓ (10) |
| Number of Heads | 1 | 1 | 10 |
| **Network Architecture** | | | |
| Policy Hidden Layers | [256, 256] | [256, 256] | [256, 256] |
| Q-Function Hidden Layers | [256, 256] | [256, 256] | [256, 256] |
| Activation Function | ReLU | ReLU | CReLU |
| **PAC-Bayes Specific (PB-SAC Only)** | | | |
| KL Coefficient ($\beta$) | 1.0 | – | – |
| Failure Probability ($\delta$) | 0.1 | – | – |
| Initial Std Dev | 0.02 | – | – |
| PB Update Frequency | 20,000 | – | – |
| Actor Freeze Frequency | 1,000 | – | – |
| Rollout Trajectories | 50,000 | – | – |
| Rollout Steps per Trajectory | 25 | – | – |
| **PBAC Specific** | | | |
| Bootstrap Rate | – | – | 0.05 |
| Posterior Sampling Rate | – | – | 5 |
| Prior Scaling | – | – | 5.0 |

