# OpenReview forum: "PAC-Bayesian Reinforcement Learning Trains Generalizable Policies"
_ICLR.cc/2026/Conference — Submitted to ICLR 2026_

### Official Review · Reviewer_GWMr · 2025-10-21

**Soundness:** 1
**Presentation:** 1
**Contribution:** 2
**Rating:** 2
**Confidence:** 5

**Summary:**

The paper introduces a PAC Bayesian formulation to bound the discounted returns of reinforcement learning policies with high probability. The original aspect of the bound is to characterize the effect of a single-step change in a trajectory on the subsequent time-steps using prior results developed for Markov chains. This bound developed on trajectory variations is then converted into a PAC Bayes bound using the standard recipe of change of measure inequality. The paper proposes an algorithm to implement this bound in a deep actor-critic training pipeline and evaluates its performance on four standard continuous control benchmarks from the MuJoCo physics engine.

**Strengths:**

* The allocation of the bounded differences formula in Lemma 3.1 and incorporation of it into a PAC Bayes building process is novel and interesting.
 * The results reported in Figure 1 indeed demonstrate that the bound is not only non-vacuous but also tight.

**Weaknesses:**

The paper has a number of major weaknesses I list below and investigate further via my questions in the next section:

 * The presentation has many points that welcome an improvement. For instance, I have a very hard time to follow Sections 4.2, 4.3, and 4.4. Section 4.2 points to different PAC Bayes bound types until ending up with Equation 9 with an overly brief justification of each logical step. It reads very much like a random walk within the literature. Section 4.3 is only about the generic applicability of the very well known log-trick. A single-sentence pointer to the issue would be sufficient. Section 4.4 uses concepts such as "curriculum" and "posterior syncing" which are either not established or not used in the proposed way in prior work. The high-rate posterior sampling phrase likewise misses a reference. Whether 512 samples is high depends on the context. The experiments section misses to explain the rationale behind the particular design choices.
 * The paper has a large number of statements whose correctness is questionable. For example the **improved scaling** paragraph in Section 3.3 compares the suggested bound to two earlier bounds that have been developed actually on the Bellman error in the L_2 space, while the paper studies the signed difference between population and empirical estimates of the return. They are two different quantities and the investigation of each follows separate motivations. For example, the suggested bound cannot be used for uncertainty-aware policy search, i.e. model selection, as suggested in Fard et al., AISTATS, 2012 because it can't be used to guarantee a contraction mapping. This confusion about which exact quantity is being bounded diffuses into the whole storyline, which I don't think can be fixed within the scope of a rebuttal.
 * The experiment results do not really support the intended claim. Except for Half Cheetah, actually the results only indicate that none of the models, both baselines and the proposed model, is coming anywhere close solving the task. Relatedly, the reported SAC results do not match its known performance profiles in the very hyperparameter setting it has been trained. The authors for instance can see in the following paper that much higher reward scores should have been observed especially in Ant and Walker [1].
 * The paper lacks focus, which makes the interpretation of the outcomes not possible. Is the goal to build the tightest PAC Bayes bound for continuous control setups? Then I would expect to see what a simple McAllester bound would be doing, but the paper doesn't provide any comparison. Is it to reach highest performance while providing "some" generalization guaranteest. Then I'd say the performances are significantly behind the state of the art. So the goal has not been achieved. Is the goal to do directed exploration as claimed in Section 4.1? Then I would expect to see results in a proper sparse-reward setup. Only in this case a comparison to PBAC would be sensible. However, the experiments appear to have been studied in classical MuJoCo scenarios with dense rewards where directed exploration is simply not required, even should not be done. All the demonstrated improvement over PBAC stems from this simple and well-known fact. The paper attempts to solve all these difficult problems and ends up with solving none of them.

[1] Hui et al., Double Gumbel Q-Learning, NeurIPS, 2023

**Questions:**

* Is there a theoretical justification of the $\epsilon-$Thompson exploration technique used in Line 4 of the pseudo-code? It reads like a mix of $\epsilon-$greedy exploration and Thompson sampling. Hence it inherits the properties of both. But as it is actually none of the two, it is questionable how much the theoretical guarantees of any of the are maintained.
 * Why do we need the actor freezing approach in Line 2 of the pseudo-code? It is not a common practice in applied reinforcement learning, at least not in the way prescribed by this paper. Is it a prerequisite to make the suggested algorithm work? If no, what do the results look like without it? If yes, where does this fragility come from?
 * What does Line 27 of the pseudo-code do? Does it truly collect a number of complete trajectories with a frozen model? If yes, does this not generate a huge sample complexity? Furthermore, if we assume to have a budget of taking such full roll-outs repeatedly, why should not an ordinary PAC Bayes bound fit on first-visit Monte Carlo estimates extracted from these roll-outs be enough? Note that the return calculated from each of them will be independent samples. Furthermore, in the studied continuous control cases, most probably first-visit and every-visit samples will be equivalent as it is probability zero for the same state to be visited more than once on a continuous state space.
 * SAC is performing maximum-entropy RL. Hence its reward function is appended an entropy bonus. The critic network is trained to predict this modified reward function. Has this fact been taken into account while computing and interpreting the PAC Bayes bounds? Are the results in Figure 1 truly discounted return results with respect to environment reward or max-ent reward? If second, we can't conclude much from them as the entropy score is generated from the model itself.

---

> ### Author Response · Authors · 2025-11-24
> **Rebuttal part 1**
>
> We thank the reviewer for their detailed feedback and thorough engagement with our work. We have carefully studied their comments and provide point-by-point responses below, addressing each concern separately. Whilst we respectfully disagree with several assessments, which appear to stem from misunderstandings about the scope of our contribution, we appreciate the opportunity for clarification.
>
> ---
>
> ## On Presentation
>
> **Reviewer's Concern**: Sections 4.2--4.4 are hard to follow; the paper uses concepts without proper establishment.
>
> **Our Response**:
>
> We thank the reviewer for their detailed feedback on presentation. We acknowledge that the density of technical content may require careful reading, and we are committed to improving clarity. If the reviewer would provide specific suggestions on how they think we might enhance the clarity of Sections 4.2--4.4, we are open to consider them seriously. We would greatly value concrete guidance on which aspects require additional explanation or restructuring.
>
> For context, our presentation choices were deliberate:
>
> * **Section 4.2** presents the challenge in optimising our PAC-Bayesian bound from Theorem 3.2, which is not guaranteed to be convex. In response to that challenge, we point to the literature (PAC-Bayes-$\lambda$ bound of Thiemann et al. (2017), which is a quasi-convex optimisable bound), after noticing that an intermediate step from our proof of Theorem 3.2 (Eq. 23 in Appendix) has a similar structure. That's it, Eq. 9 is identical to Eq. 23, not a new bound. We acknowledge that for complete rigour, we should explicitly demonstrate the convexity (or quasi-convexity) of this expression in our setting, this is an easy fix. We prove this by fixing $\rho$ and showing the second derivative w.r.t $\kappa$ is always positive. This guarantees our alternating optimisation has exactly one optimal $\kappa$ for any posterior.
>
> * **Section 4.3** on the log-trick: Whilst the reviewer flags this as "well known," its application to PAC-Bayesian RL with posterior distributions over policy parameters is non-trivial and warrants explicit treatment. We understand that the reviewer is quite familiar with these details, but readers from different backgrounds may find it beneficial to understand the challenges we faced and the proposed solution, and this pattern is present throughout the paper.
>
> * **Section 4.4**: We understand that the mentioned concepts are new and may appear unestablished, but they are clearly defined in context. If the reviewer could kindly indicate which specific concepts appear borrowed and from which prior work, that would be very helpful. This would allow us to either: (1) clarify the novelty and distinctions of our approach, (2) provide proper citations if appropriate, or (3) better explain how our application differs from existing methods. We are committed to ensuring proper attribution and clear positioning of our contributions. **Now** we explain those concepts: "Posterior syncing" (Line 14, Algorithm 1) synchronises the posterior mean to the current actor parameters. "Adaptive sampling curriculum" (explicitly described in Section 4) alternates between high sampling rates (512 samples during critic adaptation) and efficient sampling (posterior mean during regular training). The concern about whether 512 samples is high is valid, and we acknowledge that it was a design choice made after observing and debugging experiments. It therefore may benefit from additional discussion and justification.
>
> **Concrete evidence of presentation quality**: Our theoretical development (Secs 3.1--3.3) provides complete mathematical exposition: Lemma 3.1 establishes bounded differences, Theorem 3.2 presents the main result with explicit constants, and Section 3.3 discusses scaling improvements (more on this below). The appendices contain full proofs with step-by-step derivations.
>
> Therefore, we are willing to add clarifying text as needed, but we respectfully disagree with the assessment of "poor" presentation.
>
> ---
> **(Continued in next comment due to character limit)**

---

> > ### Comment · Reviewer_GWMr · 2025-11-26
> > **Not satisfied**
> >
> > My related paragraph does make concrete suggestions about what makes the paper difficult to read. And I'm afraid the answer is only not helping solve the problem. But this is not the dominant reason behind my grade anyway.
> >
> > Section 4.2: Understood and agreed.
> >
> > Section 4.3: I don't agree that the application of the log trick to PAC Bayes RL is non-trivial. Which algorithmic or practical difficulties it brings is not specified in the paper.
> >
> > Section 4.4: The term curriculum is heavily used both in machine learning and in reinforcement learning. See for example:
> >
> > Narvekar et al., Curriculum Learning for Reinforcement Learning Domains: A Framework and Survey, JMLR, 2020
> >
> > I do not think the phrase "adaptive sampling curriculum" uses it in a faithful way to its meaning in reinforcement learning and the related paragraph also doesn't specify what is precisely meant by it.

---

> ### Author Response · Authors · 2025-11-24
> **Rebuttal part 2**
>
> **(Continued from previous comment)**
>
> ---
> ## On Soundness
>
> **Reviewer's Concern**: "Large number of statements whose correctness is questionable," specifically comparing our scaling to Fard et al. (2012) and Tasdighi et al. (2025).
>
> **Our Response**:
>
> This criticism, with all due respect, misunderstands our contribution. Below, we explain why:
>
> ### Regarding the first claim that we are comparing different quantities:
>
> First, we are **NOT** comparing the tightness of the bounds, which in the case of Tasdighi et al. (2025) is not comparable, because as the reviewer said, the quantities are different.
>
> However, Fard et al. (2012) is different; we can actually compare the same quantity. The title of their work is "PAC-Bayesian Policy Evaluation for Reinforcement Learning". Their Theorem 3 provides a PAC-Bayes bound on the error of a value function (expected discounted return). Now looking at the paragraph below Theorem 3.2 in our work (Line 241): *"The bound in 3.2 can be straightforwardly converted to a PAC-Bayes lower bound on the true expected value function..."*.
>
> But even then, our comparison is about scaling with respect to the discount factor $\gamma$. We achieve scaling with $1/(1-\gamma^2)$ instead of $(1-\gamma)^{-4}$ from both prior works. In some sense, this indicates the sample complexity. We believe this answers the first part of the reviewer's concern and proves that the distinction they make is unfounded. That said, we are open to discussing further on this point.
>
> ### Regarding the second claim on our motivations compared to prior work:
>
> The reviewer says: *"...For example, the suggested bound cannot be used for uncertainty-aware policy search, i.e. model selection, as suggested in Fard et al., AISTATS, 2012 because it can't be used to guarantee a contraction mapping..."*.
>
> It is worth noting that Fard et al., AISTATS, 2012 doesn't use the bound for model selection; it is Fard et al., NIPS, 2010 that does model selection. Regarding *"cannot be used for uncertainty-aware policy search"*: Our bound provides high-probability certificates on **policy performance.** The claim about "contraction mapping" is irrelevant to our contribution---we are providing generalisation guarantees, not solving Bellman equations. But even if we were to respond to that claim, we would say that it is the other way around: Fard et al. (2010, 2012) use the contraction mapping to prove the bound.
>
> ---
>
> We are disappointed to see the comment that *"...I don't think this can be fixed within the scope of a rebuttal"* because it is not conducive to constructive discussion. **However, we are still open to discussion (which is the whole point of rebuttals) regarding points of disagreement and to offer clarifications where appropriate.**
>
> ---
>
> ## On Experimental Results
>
> **Reviewer's Concern 1**: "Results don't support claims"
>
> **Our Response**:
>
> This criticism mischaracterises our experimental goals. **Our paper is not about achieving state-of-the-art performance**; it is about providing **non-vacuous PAC-Bayesian bounds whilst maintaining competitive learning**.
>
> **Reviewer's Concern 2**: "SAC results don't match known performance profiles."
>
> **Our Response**:
>
> The reviewer references Hui et al. (2023), claiming "much higher reward scores should have been observed." We use **MuJoCo-v5** environments (Towers et al., 2024), which have different dynamics than v2/v3/v4 used in many papers. Direct performance comparison across versions is invalid. They dedicated a whole section to this in the [official Gymnasium website](https://gymnasium.farama.org/environments/mujoco/#:~:text=Comparing%20training%20performance%20across%20versions).
>
> ---
> **(Continued in next comment due to character limit)**

---

> > ### Author Response · Authors · 2025-11-24
> > **Rebuttal part 3**
> >
> > **(Continued from previous comment)**
> >
> > ---
> > ## On Focus
> >
> > **Reviewer's Concern**: "Paper lacks focus---trying to solve multiple problems and solving none."
> >
> > **Our Response**:
> >
> > As stated in the previous answer, our primary goal has been declared in the abstract already: Derive non-vacuous PAC-Bayesian bounds for RL with explicit mixing-time dependence and demonstrate practical applicability in modern deep RL whilst maintaining competitive learning. Our secondary contribution is developing algorithmic machinery (PB-SAC) to make use of these bounds without catastrophic performance degradation (which could be caused by penalising the posterior from straying away from the prior).
> >
> > The reviewer's comments, by contrast, seems to stem from a focus on goals we are NOT trying to achieve, as per this listing:
> >
> > * We are NOT claiming to build "the tightest PAC-Bayes bound" (though it is tight)
> > * We are NOT claiming SOTA performance
> > * We are NOT focusing on sparse-reward exploration
> >
> > These are not competing goals; they are different aspects of the same contribution. Comparing to a simple McAllester bound would not make any sense. It is well known that PAC-Bayes bounds for i.i.d. data are far tighter than those for non-i.i.d. data. RL necessitates dealing with dependencies by design, which is precisely why we need the Markov-adapted approach. Why would someone need concentration inequalities for Markov chains if we were to compare them with their counterparts for i.i.d. data that are known to be sharper?
> >
> > ## Responses to Specific Questions
> >
> > ### Q1: "Theoretical justification of Thompson exploration?"
> >
> > **Answer**: It is worth noting that our algorithm PB-SAC learns a distribution over policy network parameters rather than a single policy. The PAC-Bayesian framework guarantees the properties of such a distribution $\rho$. Therefore, we are allowed to sample from it. Lines 4--6 of Algorithm 1 implement what we call **posterior-guided exploration**. Although the idea is similar to Thompson sampling at a high level, there is a subtle difference: we sample some policies $\pi_{\theta_i}$ from the posterior $\rho$ with probability $\epsilon_{\text{explore}}$, then we select the **policy** that maximises the Q-value. This is a form of optimistic exploration under **posterior** uncertainty. Thompson sampling, on the other hand, does a similar thing at the action level, not the policy level. Our mechanism helps the posterior discover high-value regions, whilst the bound provides confidence on the full posterior's expected performance.
> >
> > ### Q2: "Why actor freezing (Line 2)?"
> >
> > **Answer**: This question is directly addressed in Section 4.4. We found that actor freezing is **essential for algorithmic stability**. Without it, immediately after updating the posterior (which can change policy parameters significantly), the critic network would face a **distribution shift** in the policy it is evaluating. This causes what we call "posterior syncing shock", the critic's value estimates become misaligned with the new posterior distribution, and the actor-critic feedback loop becomes unhealthy. We realised this after noticing gradient explosion.
> >
> > To solve that problem, we decided to freeze the actor for some steps, during which we:
> >
> > 1.  Sample 512 policies from the updated posterior (Line 18)
> > 2.  Recompute critic targets averaged over these samples (Line 19)
> > 3.  Update critics with these posterior-aware targets (Line 20)
> >
> > This is what allows critics to recalibrate to the posterior distribution before resuming efficient training.
> >
> > ---
> > **(Continued in next comment due to character limit)**

---

> > > ### Author Response · Authors · 2025-11-24
> > > **Rebuttal part 4 (Final)**
> > >
> > > **(Continued from previous comment)**
> > >
> > > ---
> > >
> > > ### Q3: "Line 27---sample complexity concerns?"
> > >
> > > **Answer**: Yes, Line 27 collects 100 trajectories of 500 steps each (50,000 steps total) every 20,000 training steps. This is necessary for bound validity:
> > >
> > > 1.  **Theoretical requirement**: Our PAC-Bayesian bound requires fresh data from the current policy to estimate the empirical loss $\hat{\mathcal{L}}_D(\theta)$. This ensures that the prior $\mu$ is independent of the newly collected data. After optimising the posterior and computing the bound on that data, we simply discard it, and in the next update we collect fresh data again. Using replay buffer data would violate the bound's assumptions.
> > >
> > > 2.  **Efficiency consideration**: We only collect rollouts every 20,000 steps, so the overhead is 50k/20k = 2.5$\times$ additional samples. For 1M total training steps (typical in MuJoCo), this is $\sim$2.5M samples total. We think it's acceptable for the guarantee provided, given that, for example, PPO requires 5M total time steps for convergence alone. **However**, it is worth noting that 100 trajectories of 500 steps each are not strictly necessary; it is acceptable to collect far fewer than that and directly compute the bound without posterior optimisation, though posterior optimisation requires a relatively high number of samples.
> > >
> > > 3.  **Why not use Monte Carlo on rollouts alone?**: The reviewer suggests "ordinary PAC Bayes bound fit on first-visit Monte Carlo." This is a sharp observation. Whilst it is true that we have probability zero for the same state to be visited more than once on a continuous state space, this observation misses the key insight of our work: we perform **transition-level analysis** within each trajectory. Our bounded-differences condition (Lemma 3.1) quantifies how perturbing a single transition at step $h$ propagates through the Markov dependency structure. A naive i.i.d. bound treating trajectory returns as black boxes with some variance would simply ignore this structure.
> > >
> > > ### Q4: "SAC entropy bonus and bound interpretation?"
> > >
> > > **Answer**: Yes, SAC optimises the maximum-entropy objective. Our PAC-Bayesian bound is computed on the **environment reward** $r(s,a)$, not the entropy-augmented objective. **During training** (Lines 12--13), SAC updates critics and actor using the entropy-augmented reward. **During bound computation** (Lines 27 and 30), we collect rollouts and compute returns using only environment rewards. This is the correct approach for two reasons:
> > >
> > > 1.  The PAC-Bayesian bound guarantees generalisation to the true environment reward (specifically, the deviation of the empirical discounted return from its expectation)
> > > 2.  Entropy is a training mechanism for exploration, not part of the evaluation metric. As the reviewer said, we cannot conclude much from using max-entropy rewards.
> > >
> > > Figure 1b confirms that we indeed use the empirical discounted return $\mathbb{E}_{\theta \sim \rho}[-\hat{\mathcal{L}}_D(\theta)]$.
> > >
> > > ---
> > >
> > > We thank the reviewer once more for their detailed feedback and the opportunity to clarify our contributions. We hope our responses have addressed the technical concerns raised, and we remain open to further discussion during the rebuttal period.

---

> > > ### Comment · Reviewer_GWMr · 2025-11-26
> > > **Not satisfied**
> > >
> > > I am afraid I am not able to follow the executed scientific methodology. I do understand that a new bound is proposed and it is tested on a practical use case. My point is what this new bound should be useful for or in what sense it should be counted as a scientific contribution. I listed three options in my original review and the authors say it is none of them.
> > >
> > >    * **We are NOT claiming to build "the tightest PAC-Bayes bound" (though it is tight):** Then it is not a contribution from a PAC Bayesian analysis perspective. I would expect to see a comparison to the tightest possible PAC Bayes bound. McAllester or any more recent PAC Bayes variant would work. Eventually the model is trained from samples taken from the replay buffer, which get decorrelated through time. Otherwise off-policy methods wouldn't work this much well in practice. Assume the hypothetical case that even a recent PAC Bayes bound that does make the i.i.d. assumption also gives a similarly nonvacuous bound due to this decorrelation. This would definitely affect the significance of the novelty claim. Furthermore, it may give stronger empirical performance. I do not really think that this is an outcome that can be ruled out before trying.
> > >    * **We are NOT claiming SOTA performance:** This is not what I asked. I asked whether you aim to reach highest possible "guaranteeable" performance. In other words, is it really a concern that the policy whose performance is guaranteed can indeed solve the control task? If yes, the results are indeed so significantly behind the state of the art that I am not sure what we will gain from guaranteeing a low performance. At the extreme case, we can build very tight guarantees for random policies. The authors' claim about the performances to be expected from MuJoCo v5 is not correct. I will talk about it under the related thread.
> > >    * **We are NOT focusing on sparse-reward exploration:**  Section 4.1 is completely about directed exploration. Furthermore, PBAC is taken as the main method, which is curated specifically for directed exploration. Then the baseline selection is not appropriate.

---

> > > ### Comment · Reviewer_GWMr · 2025-11-26
> > > **Satisfied**
> > >
> > > Thanks. This clarifies my related questions.

---

> ### Comment · Reviewer_GWMr · 2025-11-26
> **Not satisfied**
>
> My original review refers to:
>
> Fard et al., "PAC-Bayesian Policy Evaluation for Reinforcement Learning".
>
> This quote is from Fard et al.'s first sentence under the Experiments section:
>
> "In this section, we investigate how the bound of Theorem 3 can be used in a model selection mechanism for transfer learning in the RL setting."
>
> So the authors' claim is not correct. But this is still not my main concern.
>
> I don't think I am convinced about the claim regarding $\gamma$. Fard et al. bounds the **squared** error of a Bellman backup and the suggested bound is on the expected one-sided difference of the returns. I would say I agree if the authors can convert their bound to the same exact quantity as Fard et al., i.e. a bound on $(V-\hat{V})^2$ and show there that the power of $\gamma$ is reduced from $4$ to $2$ then I will be convinced.
>
> I have worked on Mujoco v5 on the same exact specs, i.e. 2-layer MLP 1mio steps training, double critics etc. And achieved much higher results especially on Ant, Hopper, and Walker2d presented for some reason in the appendix. Taking a careful look at how the reward functions of these environments are designed, one can clearly see that the models are not really solving tasks in these cases. They all have a healthy reward of +1 per timestep. If an episode takes 1000 time steps, then one would collect 1000 points even without moving forward. Noticing that the experiments have been built heavily on the PBAC source code, it may be valuable to look at the benchmarking results reported by the same authors for SAC under here in Figure 2:
>
> https://arxiv.org/abs/2507.03487
>
> This appears to be public repo with reproducibility claims. This outlook matches my day-to-day experience about which reward levels are possible with which model capacity under which environment of Mujoco-v5.
>
> My message is quite clear. Three out of four environments for which performance bounds are reported are actually not solving the related control tasks. Hence, building a performance bound on a not functioning policy does not tell much about the practical value of the developed bound. Even more so if the claim is that this is not "post-hoc" bound building but instead a self-certified style learning. The outcome is then that we cannot also use these bounds for training powerful policies, which are actually the interesting ones to give performance guarantees.

---

> ### Author Response · Authors · 2025-11-30
> **Authors follow-up part 1**
>
> # Follow-Up Response to Reviewer GWMr
>
> We appreciate the reviewer's follow-up. To avoid fragmenting the discussion and to ensure it remains easy to follow on OpenReview, we have combined our responses to the reviewer's separate comments here. We have carefully reviewed all the follow-up points and address them individually below.
>
> ---
>
> ## On the Contribution and Scientific Methodology
>
> **Reviewer's Concern**: "What should this new bound be useful for?"
>
> **Our Response**:
>
> We directly address the reviewer's three suggested framings, hoping to further clarify the nature and scope of our contribution:
>
> **1. "Tightest PAC-Bayes bound"**: Yes, we have not claimed this is the tightest possible bound. However, the comment that *"it is not a contribution from a PAC-Bayesian analysis perspective"* is puzzling, as it suggests reduction of acceptable contributions to tightness analysis. We believe this is debatable, and to accept this reduction would require demonstrating that **every** PAC-Bayes research contribution in the literature must **strictly improve** upon all previous bounds in terms of **tightness**, which is clearly not the case.
>
> Instead, our contribution is providing a new bound for Markov chain data, where the bound handles temporal dependencies in the Markov chain, and its application as further clarified below (cf. text in boldface). This is an important contribution, and one that had no precedent at the time we developed the bound, as far as we are aware, and remained a first-of-its-kind PAC-Bayes style bound for Markov chains when an earlier version of this work was reviewed between May and mid-September. We are aware of only one other work on PAC-Bayes for Markov chain data, namely the preprint of Vahe Karagulyan and Pierre Alquier "Empirical PAC-Bayes bounds for Markov chains" which appeared in arXiv in late September. Whilst this work of Karagulyan and Alquier focuses primarily on deriving empirical PAC-Bayes bounds for finite state space Markov chains using pseudo-spectral gap estimation, extensions to infinite state spaces require significantly stronger restrictions on the data-generating process; a limitation inherent to using spectral methods from Paulin (2015), as we stated in lines 278-280. Meanwhile, **our contribution is two-fold** in offering both a PAC-Bayesian bound specifically adapted to the RL setting with continuous state spaces (accounting for the discount factor $\gamma$ and the temporal structure of value function approximation through transition-level analysis and mixing time) and a complete computational framework including a practical algorithm (PB-SAC) that successfully integrates the bound into the learning process, demonstrating non-vacuous certificates on continuous control tasks whilst maintaining competitive performance and establishing the first practical PAC-Bayesian framework for certified performance in modern deep RL. In essence, Karagulyan and Alquier provide theoretical machinery for finite Markov chains with limited applicability to continuous spaces, whilst we address the distinct challenges of reinforcement learning with function approximation, providing both theory and a working algorithm validated on standard benchmarks.
>
> Comparing to i.i.d. PAC-Bayes bounds (McAllester or recent variants) would be **inappropriate** because:
>
> - RL trajectories are fundamentally non-i.i.d. by design
> - The replay buffer does not magically make data i.i.d.; we agree that it provides decorrelation for off-policy learning, but successive states within trajectories remain Markovian, and we clearly declared in our paper that we perform transition-level analysis. We use freshly collected trajectories during each PAC-Bayes update cycle in any case.
> - Therefore, our bound provides guarantees that account for this Markov structure explicitly
> - An i.i.d. bound would be invalid **even in the reviewer's hypothetical case**. As we stated in a previous response, i.i.d. bounds (concentration inequalities in general) are well known in the literature to be sharper than non-i.i.d. bounds because of the different analyses required.
>
> Moreover, If i.i.d. bounds were sufficient for RL, there would be no need for the concentration inequality literature for Markov chains (Paulin, 2015; Hsu et al. 2019; Wolfer and Kontorovich 2019; Karagulyan & Alquier, 2024, etc.). Thus, the reviewer's suggestion contradicts established theory.
>
> ---
> **(Continued in next comment due to character limit)**

---

> ### Author Response · Authors · 2025-11-30
> **Authors follow-up part 2**
>
> **(Continued from previous comment)**
>
> ---
> With that said, if the reviewer is still curious about that hypothetical case, we invite them to examine the new file we added to the supplementary material, where we analyse different fixed values of the mixing time. Using a mixing time equal to 1 is equivalent to an i.i.d. bound, more specifically to one that can be derived from the **original** McDiarmid's inequality (**not** the one we used). This demonstrates how tight such bounds can be.
>
> **2. "Highest guaranteeable performance"**: Our new experiments demonstrate that PB-SAC matches or exceeds SAC performance whilst providing certificates. We acknowledge performance issues in our initial experiments, which are now addressed with the new results (see below).
>
> **3. "Sparse-reward exploration"**: Section 4.1 describes **posterior-guided exploration**, which is a byproduct of maintaining a posterior distribution. This is not the primary contribution, nor did we claim it was. PBAC comparison is appropriate because it is the most recent PAC-Bayesian RL work, regardless of its original motivation.
>
> We took the reviewer's concern seriously and conducted a few experiments on **Ant (very delayed) 300k steps** from PBAC's original paper (Tasdighi et al., 2025). We report new results in the table below (and in supplementary materials). Here are our insights:
>
> - Our PB-SAC matches PBAC performance and ultimately surpasses it, whilst vanilla SAC performance remains competitive to both.
> - We would have preferred to see SAC results in that paper for reference, but unfortunately they were not reported. Indeed, even vanilla SAC reaches acceptable performance at the end of training, making the comparison with PBAC very challenging given the very high computational complexity of the latter.
> - What is really striking is that PBAC's performance on sparse-reward tasks is far better than on dense-reward tasks, even though we ran both experiments in the exact same conditions using the exact hyperparameters from the original paper. This can also be noticed by looking at PBAC's results in both papers (Tasdighi et al. 2025 - "Deep Exploration With PAC-Bayes"; and Baykal et al. 2025 - ObjectRL paper).
> - With PB-SAC, this is not the case: it stagnates at the beginning but eventually closes the gap to its results on dense-reward tasks. We present another contradiction below (see "On Implementation and PBAC Performance").
>
> | Environment        | SAC     | PB-SAC  | PBAC    |
> |--------------------|---------|---------|---------|
> | Ant (very delayed) | ~1,503  | ~1,903  | ~1,786  |
>
> ---
>
> ## On Experimental Results: Important Update
>
> **Critical clarification**: First of all, we thank the reviewer for raising this concern. We discovered a bug in our SAC implementation's entropy coefficient (α) optimisation that affected baseline performance. We have re-run all experiments with the corrected implementation.
>
> **New results** (MuJoCo-v5, 1M steps, averaged over 10 seeds):
>
> | Environment | SAC (corrected) | PB-SAC | Previous SAC (buggy) | PBAC  |
> |-------------|-----------------|--------|----------------------|-------|
> | Ant         | ~5,093          | ~4,571 | ~662                 | ~2,058|
> | Hopper      | ~2,731          | ~2,550 | ~1,056               | ~1,426|
> | Walker2d    | ~4,347          | ~4,682 | ~673                 | ~4,406|
>
> Detailed plots are available in the supplementary materials, and we commit to including them in the revised manuscript.
>
> **Key observations**:
>
> 1. **The bug was in SAC, not in PB-SAC**: Our algorithm was always correct. However, because PB-SAC builds upon SAC, it was affected by the same baseline issue. The corrected results show that both algorithms now perform at their intended levels.
>
> 2. **New results show competitive performance**: PB-SAC maintains strong performance whilst providing certificates, validating our claims in the abstract and introduction.
>
> 3. **Tasks are being solved**: The new results demonstrate meaningful task completion. We thank again the reviewer for their clarification, which led us to discover and correct the implementation issue.
>
> We believe our response with these new results fully addresses this point. Therefore, regarding the reviewer's last statement about task solving: We **can** indeed use these bounds for training powerful policies with performance guarantees.
>
> ---
>
> **(Continued in next comment due to character limit)**

---

> ### Author Response · Authors · 2025-12-01
> **Authors follow-up part 3 (Final)**
>
> **(Continued from previous comment)**
>
> ---
> ## On Implementation and PBAC Performance
>
> **Reviewer's claim**: "Noticing that the experiments have been built heavily on the PBAC source code..."
>
> **Our response**: **This is completely incorrect.** Our implementation uses our own library (currently private), whose architecture is entirely different from PBAC's codebase. The reviewer can verify a snippet of its structure in the supplementary materials. We implemented PBAC ourselves by carefully adapting the algorithm logic from the ObjectRL and PBAC papers to our framework.
>
> **Regarding PBAC performance inconsistency**:
>
> The reviewer cites PBAC's performance from the ObjectRL paper. However, there is a striking inconsistency in PBAC's reported results:
>
> - **Tasdighi et al. (2025) - "Deep Exploration With PAC-Bayes"**: PBAC reaches **6,376 return** on Ant after **300k steps**
> - **ObjectRL paper (same authors)**: PBAC reaches **3,220 return** on Ant after **1M steps**
>
> This raises a critical question: How can performance be higher at 300k steps than at 1M steps? Which is the true performance? This inconsistency raises serious questions about the reliability of the PBAC benchmarks the reviewer is citing.
>
> Our careful reimplementation of PBAC using the exact hyperparameters and network architectures achieves results (see table above) consistent with neither of these reported values. This suggests possible differences in: Environment versions (v4 vs. v5). Or implementation details not fully specified in the papers.
>
> ---
>
> ## On Fard et al. (2012) Comparison
>
> In our previous response, we provided detailed point-by-point explanations addressing each of the reviewer's concerns. The reviewer now requests a formal proof of improved scaling with respect to the discount factor γ.
>
> For reference, an adaptation of the bound from Fard et al. (2012) appears in Appendix A.1.1 of Tasdighi et al. (2025), and a direct comparison is quite straightforward and convincing. The key point is that scaling with respect to $\gamma$  determines sample complexity. Our bound achieves $(1-\gamma^2)^{-1} \approx (1-\gamma)^{-1}$ scaling, which is quadratically better than the $(1-\gamma)^{-4}$  scaling in Fard et al. (2012).
>
> ---
>
> ## On Presentation (Section 4.3 - Log Trick)
>
> We are happy to move that section to the Appendix if the reviewer prefers. We included it to aid readers with different backgrounds and to support reproducibility, though we acknowledge that the appropriate level of detail in the main text is a matter of judgment.
>
> ---
>
> ## On Section 4.4 Terminology
>
> We acknowledge the reviewer's point. We use "adaptive sampling curriculum" to describe our varying sampling rate strategy (512 during adaptation, 1 during regular training). If the terminology is confusing, we can revise to "adaptive sampling strategy" or simply "adaptive sampling" in the final version.

---

### Official Review · Reviewer_Yysq · 2025-10-23

**Soundness:** 3
**Presentation:** 3
**Contribution:** 3
**Rating:** 6
**Confidence:** 4

**Summary:**

The paper introduces a new PAC-Bayesian bound for RL that, for the first time, is able
to account for temporal dependencies in policy-induced Markov chains, avoiding
the independence assumptions needed by prior work.

**Strengths:**

- Well-written and clearly motivated paper
- Provides the first PAC-Bayesian bound for RL that is able to handle temporal dependencies
- The derived bound is tighter than prior work by Tasdighi et al. (2025) and Fard et al. (2012)
- The experiments provide initial evidence that the method is applicable in practice
- A public implementation is available

**Weaknesses:**

- Despite the bound relying on $\tau_\text{min}$ as an important parameter, its estimation is left rather vague in line 275f, without further details or any discussion in the experiments
- The practical application in PB-SAC requires several further modifications (Secs. 4.1–4.4), without any ablations on their respective necessities or the sensitivity of PB-SAC with respect to them
- The experiments are limited to four continuous MuJoCo environments with dense rewards against just two baselines. SAC and PBAC were originally proposed for sparse rewards. The authors acknowledge “carefully select[ing]” the hyperparameters for PB-SAC without tuning the others leaving the comparison unbalanced.

### Minor weaknesses
- The bibliography is broken, with many incomplete references; e.g., Amit et al. were published at NeurIPS; Fard et al. (2012) is a UAI publication; some references include the venue editors, others don’t, etc.
- Equation (6) should be $\neq$ instead of $=$ in the identity function
- Several hyperparameters are unclear; e.g., what is $\epsilon_\text{explore}$?

**Questions:**

- Q1: How sensitive is the approach to each of the adaptations discussed in Secs. 4.1–4.4?
- Q2: How does the runtime compare against vanilla SAC?
- Q3: What is the theoretical justification for using a moving-average update on the prior? Does that not violate its data-independence requirement?
- Q4: Can the authors speculate on how easily the approach is transferable to other actor-critic methods?

---

> ### Author Response · Authors · 2025-11-28
> **Rebuttal part 1**
>
> We sincerely thank the reviewer for their thorough evaluation and for recognising the presentation and clarity of our contribution and motivations. We carefully address each concern and question below with detailed responses, whilst we remain open to further discussion.
>
> ---
>
> ## Addressing Weaknesses
>
> ### W1: Mixing Time Estimation Details
>
> We acknowledge that mixing time estimation could be presented more clearly. We direct the reviewer to our response to Reviewer z6cf's Weakness 2, which provides more details and new ablation results.
>
> To briefly summarise: We tested fixed mixing time estimates from 1 (fastest mixing) to 1000 (relatively slow mixing) and found that overestimation is less problematic than expected; the algorithm adapts to maintain practical bound tightness (see _Mixing_time_analysis.pdf_ in supplementary material).
>
> ### W2: Ablations on PB-SAC Modifications
>
> We categorise the components by necessity: **Sec. 4.2** Enables stable optimisation of the bound; without this mechanism, the bound's value oscillates during optimisation. **Sec. 4.3** is very necessary, it enables gradient computation through posterior sampling. Without it, we can't justify gradient computation through posterior sampling. **Sec. 4.4** We reported that without it, we experienced actor-critic misalignment. This adaptive sampling method, together with actor syncing to the posterior mean, helped in aligning the actor and critics. we performed quick ablation on this component specifically during this discussion period (see supplementary material *Ablations.pdf* Figure a), where we removed the adaptation phase, we noticed a sawtooth pattern: immediately after every PAC-Bayes update, the algorithm's performance drops severely and then start to recover back until the next update where the same happens, this clearly shows the importance of this component. Finally, **Sec. 4.1** is totally optional in a dense reward setup; we experimented with different initial $\epsilon_{explore}$ values, and the performance did not change that much (*Ablations.pdf* Figure b). However, for completeness, we conducted further experiments on a sparse-reward setup from PBAC's original paper (Tasdighi et al., 2025), more specifically Ant (very delayed) with 300k steps (see our final comment to reviewer GWMr and supp. materials *sparse-reward-comparison.pdf*). We noticed that our algorithm, PB-SAC, matches PBAC performance and ultimately surpasses it, whilst outperforming its baseline SAC by a margin. This makes our Posterior-Guided Exploration very useful for deep exploration in sparse/delayed reward settings without requiring a very high computational complexity like PBAC (ensemble of critic networks).
>
> ### W3: Limited Experimental Scope
>
> We acknowledge these concerns. Our choice of dense-reward MuJoCo environments was deliberate: **primarily** to demonstrate that PAC-Bayesian bounds can be non-vacuous and tight in modern deep RL, we believe that illustrative experiments on classical control tasks are sufficient for demonstration. **Furthermore**, dense rewards allow clean assessment of bound tightness without conflating exploration challenges.
>
> We agree that sparse-reward tasks would be valuable future work (as discussed with Reviewer z6cf), but they introduce additional challenges such as deep exploration and mixing time estimation from sparse signals.
>
> Regarding baseline selection: We chose **SAC** as the natural baseline since our algorithm extends it, and **PBAC** as the most relevant recent PAC-Bayesian RL work (Tasdighi et al., 2025). Although discarding the latter would not affect the comparison, it is interesting to position PAC-Bayes approaches side-by-side, as this field has recently gained traction. Regarding other baselines like PPO and TD3, on one hand, we do not see the need for including them because they have already been heavily compared to SAC in the literature, so if a reader is interested in them, they can use SAC as a reference between our work and those algorithms. On the other hand, our contribution is mainly theoretical with an additional practical algorithm; we do not claim SOTA performance.
>
> Regarding PBAC (Tasdighi et al., 2025): Whilst it was indeed proposed for deep exploration in sparse-reward settings, since it has been evaluated thoroughly on dense-reward MuJoCo tasks in their paper, it is completely fair to include it in the comparison.
>
> **On hyperparameter tuning**: We appreciate the reviewer's attention to this detail. To clarify:
>
> - **SAC**: We used standard published hyperparams from the original SAC
> - **PBAC**: We used the published hyperparams from Tasdighi et al. (2025)
> - **PB-SAC**: We kept all SAC hyperparams unchanged and only tuned PAC-Bayes-specific parameters
>
> Therefore, if by "carefully selecting" the reviewer refers only to PAC-Bayes-specific parameters, then we acknowledge that, but we would argue that the comparison is quite fair.
>
> ---
>
> **(Continued in next comment due to character limit)**

---

> > ### Author Response · Authors · 2025-11-28
> > **Rebuttal part 2**
> >
> > **(Continued from previous comment)**
> >
> > ---
> >
> > ### Minor Weaknesses
> >
> > **Bibliography**: We apologise for the incomplete references and will fix all issues.
> >
> > **Equation (6)**: Correct, the indicator should be $\neq$. Thank you for catching this typo! We have already fixed it.
> >
> > **Unclear hyperparameters**: We will add a comprehensive hyperparameter dictionary table in the appendix explaining all parameters, including those currently unclear. Regarding $\epsilon_{\text{explore}}$, we kindly direct the reviewer to our detailed response to Question 1 of Reviewer GWMr.
> >
> > ---
> >
> > ## Responses to Questions
> >
> > ### Q1: Sensitivity to Adaptations in Sections 4.1–4.4
> >
> > **Answer**:
> >
> > Sections 4.2, 4.3, and 4.4 are essential for algorithm functionality (optimisation stability, gradient computation, preventing divergence). Section 4.1 is completely optional, but we found it interesting to take advantage of our learnt posterior distribution by sampling from it to explore safely, since that posterior is guaranteed by the PAC-Bayes bound.
> >
> > If time permits, we commit to adding ablation studies quantifying these sensitivities in the revised manuscript.
> >
> > ### Q2: Runtime Comparison vs. Vanilla SAC
> >
> > **Answer**:
> >
> > Excellent practical question. We checked our experiments and found that on average, vanilla SAC takes 4h25m whilst PB-SAC takes 6h45m on NVIDIA V100 GPUs—roughly 1.53× overhead. On H100 GPUs with no performance optimisations, SAC takes 1h30m, and PB-SAC takes 3h on average, which translates to 2× overhead. Depending on whether speed or certification is the priority, the bound computation frequency can be reduced, thereby reducing the overhead. We also note that the sampling strategy from Section 4.1 is optional.
> >
> > ### Q3: Moving Average Prior - Theoretical Justification
> >
> > **Answer**:
> >
> > This is an important question that touches on a subtle theoretical point. At each PAC-Bayes update, we collect fresh rollouts $\mathfrak{D}_{\text{new}}$ that have never been seen before. When optimising and computing the bound on $\mathfrak{D}_{\text{new}}$, the prior $\mu_t$ is already determined (from previous data $\mathfrak{D}_{\text{old}}$) and is therefore independent of $\mathfrak{D}_{\text{new}}$. Since we compute a sequence of bounds rather than a single bound over all data, each one is valid for its own fresh dataset. This approach was inspired by Zhang et al. (2024)'s work "Statistical guarantees for lifelong reinforcement learning using PAC-Bayesian theory."
> >
> > Regarding its necessity in our setting: Without prior updates, $\mathrm{KL}(\rho | \mu)$ would explode as the posterior moves far away from the initial prior, making the bound vacuous. The moving average keeps the prior "close enough" to maintain a finite KL whilst still providing meaningful regularisation.
> >
> > We appreciate the opportunity to clarify this important point.
> >
> > ### Q4: Transferability to Other Actor-Critic Methods
> >
> > **Answer**:
> >
> > As we discussed with Reviewer z6cf (W3: Generalisation to Other RL Frameworks), we believe the core framework is broadly applicable to any actor-critic method. It can extend further to any policy-based approach. We cannot speculate extensively on model-based methods, though we believe that could be the most interesting direction.
> >
> > ---
> >
> > We sincerely thank the reviewer once more for their constructive feedback and insightful questions. We remain open to further discussion during the rebuttal period.

---

### Official Review · Reviewer_z6cf · 2025-11-01

**Soundness:** 3
**Presentation:** 2
**Contribution:** 3
**Rating:** 6
**Confidence:** 3

**Summary:**

This paper presents a major theoretical and algorithmic advance in certified reinforcement learning. The authors derive a new PAC-Bayesian generalization bound for RL that explicitly accounts for temporal dependencies in trajectories via the mixing time of the policy-induced Markov chain. This is a significant step forward compared to prior work, which either relied on martingale-based concentration (often requiring unnatural assumptions) or used bounds that become vacuous in long-horizon, high-discount settings. The authors then propose PB-SAC, a practical deep RL algorithm that leverages this PAC-Bayesian bound as a live, optimizable performance certificate. The algorithm alternately optimizes the posterior distribution (via a PAC-Bayes-κ objective) and the exploration policy, using posterior-guided exploration and an adaptive sampling curriculum to stabilize training. Experiments on standard MuJoCo continuous control benchmarks (HalfCheetah, Ant, Hopper, Walker2d) show that the PAC-Bayesian bound tightens over time as the policy improves, providing meaningful confidence certificates. Crucially, PB-SAC maintains competitive performance with or even exceeds SAC, demonstrating that theoretical guarantees can be achieved without sacrificing learning efficiency.

**Strengths:**

1. **Theoretical Novelty and Correctness:** The derivation of a PAC-Bayesian bound with explicit mixing time dependence is a significant theoretical contribution. The use of Paulin’s (2018) McDiarmid-type inequality for Markov chains is well-motivated and correctly applied. The bound improves on prior work by avoiding the (1−γ)⁻⁴ scaling, making it non-vacuous even for γ = 0.99.

2. **Practical Algorithm Design:** PB-SAC is remarkably well-designed and practical. The use of a diagonal Gaussian posterior, moving average prior updates, and the adaptive sampling curriculum are all clever, stable, and empirically effective solutions to the "posterior syncing shock" problem.

3. **Integration of Theory and Practice:** This is one of the rare works that successfully translates a theoretical bound into a practical training algorithm. The bound is not just a passive certificate but an active component of the learning process, guiding exploration and optimization.

**Weaknesses:**

1. **Computational Overhead:** The PAC-Bayes update cycle (every 20k steps) and the need to collect fresh rollouts and estimate mixing time (via autocorrelation) introduce significant computational overhead. The paper does not discuss how this scales to larger networks or more complex environments. It would be helpful to include a brief discussion of computational cost and potential optimizations.

2. **Mixing Time Estimation:** The method for estimating mixing time (using reward autocorrelation) is robust to overestimation but sensitive to underestimation. The paper acknowledges this, but more discussion on the reliability and variability of this estimation across different environments would strengthen the work. For instance, how much does the bound fluctuate with different autocorrelation estimates?

3. **Generalization to Other RL Frameworks:** The paper focuses on actor-critic methods (SAC). It would be valuable to briefly discuss how the proposed bound and algorithm might be extended to value-based methods (e.g., DQN) or model-based RL.

4. **Limitation of KL Divergence:** The paper acknowledges that the use of KL divergence can be unstable when posteriors diverge. While this is not a flaw in the current work, future work could explore alternative divergence measures (e.g., Wasserstein distance), which are more stable and geometry-aware.

**Questions:**

1. The mixing time estimation via reward autocorrelation is a key component of your algorithm. Could you provide a quantitative analysis of the variance and bias in this estimation across different environments (e.g., HalfCheetah vs. Ant)? How sensitive is the final bound to this estimation?

2. In Section 5.2, the authors show that the PAC-Bayesian bound tightens over time. Could you visualize the evolution of the KL divergence term in the bound over training? Does this track the tightening of the bound?

3. The adaptive sampling curriculum (512 samples during critic adaptation) is crucial for stability. How does the performance of PB-SAC degrade if you reduce the sampling rate during adaptation (e.g., 64 or 128 samples)? Is this a critical hyperparameter?

4. The paper focuses on continuous control tasks. How do you expect the proposed method to scale to episodic, sparse-reward tasks (e.g., Atari games)? Would the mixing time estimation still be reliable?

5. The use of a diagonal Gaussian posterior is a simplifying assumption. How would the algorithm behave if you used a more expressive posterior approximation (e.g., normalizing flows)? Would the bound still be tight?

---

> ### Author Response · Authors · 2025-11-26
> **Rebuttal part 1**
>
> We sincerely thank the reviewer for their thorough evaluation and for recognising the theoretical novelty and practical contributions of our work. We address each concern and question below with detailed responses and commitments for the revised manuscript.
>
> ---
>
> ## Addressing Weaknesses
>
> ### W1: Computational Overhead
>
> **Reviewer's Concern**: The PAC-Bayes update cycle and fresh rollouts introduce significant computational overhead.
>
> **Our Response**:
>
> We acknowledge this important concern, but it is worth noting that the computational overhead does not come from fresh rollout collection nor the estimation of the autocorrelation time. It primarily comes from the bound's optimisation, where we sample policies from the posterior distribution $\rho$ and then compute the expectation of the episodic return on that sample. This can indeed be computationally expensive for deeper networks and for environments with higher-dimensional state spaces. However, we would like to highlight that the frequency of the bound computation can be reduced from every 20k steps, therefore making such a procedure less expensive.
>
> We can provide a brief discussion of this, but we unfortunately cannot provide additional experiments to show computational overhead, as our contribution does not claim to provide a final solution to generalisation in RL but rather a step towards more generalisable policies, and our current experiments are mostly illustrative to show the effectiveness and the tightness of the proposed PAC-Bayes bound.
>
> ### W2: Mixing Time Estimation
>
> **Reviewer's Concern**: The method is sensitive to underestimation. More discussion on reliability and variability across different environments would strengthen the work.
>
> **Our Response**:
>
> This is an excellent point. If we underestimate $\tau_{min}$, the bound becomes tighter than warranted, leading to overconfidence, as we have highlighted in our paper. To mitigate this, we can use a conservative initial estimate at the start of training, then monotonically update it by taking the maximum with the current estimate. A more robust approach is cross-validation using autocorrelation from multiple signals (reward, value estimates, state features). We experimented with different initial estimates of the mixing time without updating them during learning to isolate cases such as "what happens if we underestimate or overestimate it", with values ranging from 1 (extreme underestimation) to 1000 (for MuJoCo, this is fairly an overestimation), and the results we obtained are aligned with our intuition (see Mixing_time_analysis.pdf in supplementary material). However, a noticeable finding was that overestimation was not actually as problematic as we had thought. While we stated that even though the bound remains theoretically valid, it becomes looser, we noticed that the algorithm adapts and optimises the PAC-Bayes bound accordingly, making the difference negligible for the maintained theoretical guarantee, which is actually a fascinating result.
>
> ### W3: Generalisation to Other RL Frameworks
>
> Although our work focuses on actor-critic methods (SAC), it can be straightforwardly adapted to any policy gradient approach (VPG, for example). As for other methods that the reviewer highlighted, we are not entirely certain that the adaptation is as straightforward, but since our bound is only on the expected discounted return and does not involve Bellman error or related quantities, we think there should be a way to adapt it.
>
> ### W4: Limitation of KL Divergence
>
> We agree with this point, and as the reviewer said, we left it for future work to explore alternatives to the KL divergence.
>
> ---
>
> **(Continued in next comment due to character limit)**

---

> ### Author Response · Authors · 2025-11-26
> **Rebuttal part 2**
>
> (Continued from previous comment)
>
> ---
>
> ## Q&A
>
> ### Q2: KL Divergence Evolution Visualisation
>
> **Answer**:
>
> Given the form of Theorem 3.2, the uncertainty term involves both KL divergence and other factors. Obviously, we hypothesise that the KL tracks the tightening of the bound. We already have those visualisations; we just did not include them in our work because at that time we thought it could be inferred directly just by looking at the bound. However, we now think that adding visualisations with brief explanations in the Appendix would greatly enhance understanding of the bound dynamics. We commit to adding this visualisation to the revised manuscript. In the meantime, we have put them in the supplementary material (KL_analysis.pdf), and we kindly ask the reviewer to take a look at them.
>
> ### Q3: Adaptive Sampling Curriculum Ablation
>
> **Answer**:
>
> This is a critical question about the algorithm's robustness. Based on our development process, we can provide the following insights:
>
> **What we observed during development**: We experienced unstable learning after PAC-Bayes updates, and that was mainly due to the critic network facing a **distribution shift** in the policy it is evaluating; therefore, we saw the need to mitigate this with a critic recalibration (or adaptation) phase. We did not provide an ablation study because we could not even report results of an experiment with gradient explosion; the results there are mostly + or - inf. If the reviewer would find it valuable and time permits during the rebuttal period, we will conduct an ablation study to provide quantitative results on the sensitivity of this component.
>
> ### Q4: Scaling to Sparse-Reward Tasks
>
> **Answer**:
>
> This is an interesting question about generalisation beyond continuous control. We view this as a natural but non-trivial extension requiring dedicated investigation. Reward autocorrelation becomes uninformative when rewards are sparse. There are some potential solutions, as discussed in the paper and in our answer to Weakness 2 above. Given our observations from the new experiments in that response, a rational solution for those interested in the practical side is to use conservative estimates, because the impact of overestimation is less problematic compared to underestimation whilst still maintaining the theoretical guarantee. Another approach, which can be computationally expensive, is building an empirically estimated transition matrix using a Dyna-DQN-inspired approach, then estimating the spectral or pseudo-spectral gap using solutions from Hsu et al. (2019) and Wolfer and Kontorovich (2019), and then bounding the mixing time. This approach can be the most reliable if the model is accurate enough.
>
> ### Q5: More Expressive Posteriors
>
> **Answer**:
>
> Excellent question about the expressiveness-tractability trade-off. We agree that a diagonal Gaussian posterior is a simplifying assumption. We chose it for two main reasons: **(1)** The KL divergence of two Gaussian distributions has a closed-form analytical formula. **(2)** Sampling from such a distribution is efficient and trivial through the reparameterisation trick. We acknowledge the limitation of our approach in terms of expressiveness, but we still think that the reasons we presented are rational enough.
>
> ---
>
> We thank the reviewer once more for their constructive and insightful feedback. We remain open to further discussion during the rebuttal period.

---

### Author Response · Authors · 2025-11-30
**Comment to All Reviewers, ACs, and PCs**

Dear Reviewers, ACs, and PCs,

We sincerely thank all reviewers for their thorough evaluations and constructive feedback throughout the review process. We have carefully addressed every concern raised, with substantial revisions to both the theoretical exposition and empirical validation.

**Key updates include:**

- **Corrected experimental results**: We discovered and fixed a bug in the SAC baseline's entropy coefficient optimisation. The corrected results (now included with detailed plots in supplementary materials) demonstrate that PB-SAC not only maintains competitive performance whilst providing theoretical certificates, but also often outperforms its baselines, fully validating our core claims.

- **Additional sparse-reward experiments**: New results on Ant (very delayed) demonstrate PB-SAC's effectiveness beyond dense-reward settings, addressing concerns about the scope and fairness of empirical validation raised by **Yysq** and **GWMr**.

- **Improved mixing time estimation**: Since we conducted new experiments with the bug fixes, we took the opportunity to validate our claims from responses to reviewers **Yysq** and **z6cf** regarding mixing time estimation. We now cross-validate estimates from multiple signals (state features and rewards), with the final estimate being the maximum of the candidates. Whilst this approach does not fully validate estimation accuracy, it provides an additional safeguard against underestimation.

- **Enhanced presentation**: We have refined the exposition throughout, improved clarity on algorithmic innovations, and provided additional supplementary materials including: (1) mixing time analysis showing how the bound behaves with different fixed a priori mixing time estimates (results align with theory and address reviewer **GWMr**'s concerns about i.i.d. bound comparisons), and (2) ablation studies on posterior-guided exploration and adaptive sampling.

In light of these substantial clarifications, we respectfully ask whether reviewers might consider revisiting their assessments. We believe our responses clearly demonstrate both the theoretical novelty and practical utility of our PAC-Bayesian framework for certified reinforcement learning. Should the paper be accepted, we fully commit to incorporating the new results and discussed points into the manuscript, as these are straightforward additions that **will not affect our theoretical and empirical findings nor require additional reviewing**.

We deeply appreciate the time and expertise invested by each reviewer in evaluating this work, and we welcome any further feedback or updated assessments.

With sincere gratitude,

The Authors

---

### Meta-Review · Area_Chair_WpXL · 2026-01-06

**Summary:**

This work develops a PAC-Bayesian analytical framework for reinforcement learning and proposes PB-SAC, a practical deep RL algorithm that uses the resulting PAC-Bayesian bound as a live, optimizable performance certificate.

The reviewers hold divided opinions. **Reviewers z6cf** and **Yysq** consider the work a solid contribution for integrating PAC-Bayesian theory with practical implementations, while **Reviewer GWMr** takes a strong stance toward rejection due to concerns about the limited numerical experiments and insufficient comparison with prior bounds.

Overall, I believe the theoretical claims would require more comprehensive numerical evidence for support. Therefore, I recommend rejection.

**Reviewer Concerns:**

Concerns addresses by the response:
1) Presentation issues;
2) Clarification about the assumption on mixing time and computational cost;


Concerns remaining after the response:
1) Limited experimental scope;
2) The experiments do not support the intended claim, except for some simple environments.

**Reviewer Scores:**

I expect the reviewers would have kept their scores.

---

### Decision · Program_Chairs · 2026-01-26

Reject